# A TOPBP1 allele causing male infertility uncouples XY silencing dynamics from sex body formation

Carolline Ascenção[1†], Jennie R Sims[1†], Alexis Dziubek[1], William Comstock[1], Elizabeth A Fogarty[1], Jumana Badar[1], Raimundo Freire[2,3,4], Andrew Grimson[1], Robert S Weiss[5], Paula E Cohen[5]*, Marcus B Smolka[1]*

[1]Department of Molecular Biology and Genetics, Weill Institute for Cell and Molecular Biology, Cornell University, Ithaca, United States; [2]Fundación Canaria del Instituto de Investigación Sanitaria de Canarias (FIISC), Unidad de Investigación, Hospital Universitario de Canarias, Santa Cruz de Tenerife, Spain; [3]Instituto de Tecnologías Biomédicas, Universidad de La Laguna, La Laguna, Spain; [4]Universidad Fernando Pessoa Canarias, Las Palmas de Gran Canaria, Spain; [5]Department of Biomedical Sciences, Cornell University, Ithaca, United States

*For correspondence:
paula.cohen@cornell.edu (PEC);
mbs266@cornell.edu (MBS)

[†]These authors contributed equally to this work

Competing interest: The authors declare that no competing interests exist.

**Abstract** Meiotic sex chromosome inactivation (MSCI) is a critical feature of meiotic prophase I progression in males. While the ATR kinase and its activator TOPBP1 are key drivers of MSCI within the specialized sex body (SB) domain of the nucleus, how they promote silencing remains unclear given their multifaceted meiotic functions that also include DNA repair, chromosome synapsis, and SB formation. Here we report a novel mutant mouse harboring mutations in the TOPBP1-BRCT5 domain. *Topbp1*[B5/B5] males are infertile, with impaired MSCI despite displaying grossly normal events of early prophase I, including synapsis and SB formation. Specific ATR-dependent events are disrupted, including phosphorylation and localization of the RNA:DNA helicase Senataxin. *Topbp1*[B5/B5] spermatocytes initiate, but cannot maintain ongoing, MSCI. These findings reveal a non-canonical role for the ATR-TOPBP1 signaling axis in MSCI dynamics at advanced stages in pachynema and establish the first mouse mutant that separates ATR signaling and MSCI from SB formation.

## eLife assessment

This **important** study reports a new mutant mouse line with compromised function of a DNA damage response protein. The evidence supporting the conclusion that the mutants display defective maintenance of meiotic sex chromosome inactivation is **solid**. This work is of interest to biomedical researchers working on meiosis and meiotic sex chromosome inactivation.

## Introduction

During prophase I, the SPO11 topoisomerase-like enzyme and its cofactors induce programmed DNA double-strand breaks (DSBs) that are then recognized by the DNA damage response (DDR) machinery to promote recombination between homologous chromosomes (*Handel and Schimenti, 2010*; *Joshi et al., 2015*; *Keeney et al., 1997*; *Subramanian and Hochwagen, 2014*). Proper chromosome synapsis achieved through the formation of the proteinaceous synaptonemal complex, together with homologous recombination (HR)-mediated DNA repair (*Pereira et al., 2020*), is critical for the formation of crossovers that ensure the correct segregation of chromosomes and the formation of healthy and genetically diverse haploid gametes (*Gray and Cohen, 2016*). Chromosomes that fail to synapse

as prophase I progresses trigger a process referred to as meiotic silencing of unsynapsed chromatin (MSUC) (*Abe et al., 2020*; *Burgoyne et al., 2009*; *Turner, 2015*; *Turner et al., 2006*) to silence genes at unsynapsed regions. In the heterogametic sex (male mammals), the X and Y chromosomes pose a unique challenge for meiotic progression since they bear homology only at the pseudoauto-somal region (PAR). The non-homologous arms of the sex chromosomes remain unsynapsed and must therefore undergo a sex-chromosome specific manifestation of MSUC, termed meiotic sex chromo-some inactivation (MSCI) (*Alavattam et al., 2018*; *Royo et al., 2010*; *Turner, 2007*; *Turner et al., 2006*). MSCI is critical for normal prophase I progression through two complementary mechanisms, the silencing of toxic Y-linked genes, such as *Zfy1* and *Zfy2*, that enforce the pachytene checkpoint (*Royo et al., 2010*; *Vernet et al., 2016*) and through the accumulation of the DDR machinery at the X and Y chromosomes, away from the autosomes, during early pachynema (*Abe et al., 2020*).

The apical serine-threonine kinase ataxia telangiectasia mutated and Rad-3 related (ATR) is a master regulator of DNA repair, checkpoints, and silencing during prophase I in spermatocytes. In response to DSBs and asynapsis, ATR activation promotes a range of downstream effects, including recombinational DNA repair, crossing over, chromosome synapsis, cell cycle arrest, and potentially apoptosis (*Abe et al., 2022*; *Pacheco et al., 2018*; *Pereira et al., 2020*; *Royo et al., 2013*; *Widger et al., 2018*). During leptonema and zygonema, shortly after DSB formation, ATR and its cofactor ATRIP are recruited to RPA-coated regions of single-stranded DNA (ssDNA) that accumulate upon 5′–3′ resection of both ends of DSBs (*Cimprich and Cortez, 2008*; *Fanning et al., 2006*). ATR activa-tion requires recruitment of TOPBP1 (topoisomerase 2 binding protein 1), a multi-BRCT (BRCA C-ter-minus motif) domain protein that stimulates ATR kinase activity through its ATR-activation domain (AAD) (*Cimprich and Cortez, 2008*; *Kumagai et al., 2006*; *Mordes et al., 2008*; *Zhou et al., 2013*). In addition to activating ATR, TOPBP1 also serves as a scaffold for a range of DDR factors, inter-acting with, and often recruiting them via its multiple BRCT domains (*Bigot et al., 2019*; *Blackford et al., 2015*; *Cescutti et al., 2010*; *Delacroix et al., 2007*; *Leimbacher et al., 2019*; *Leung et al., 2011*; *Liu et al., 2017*; *Pereira et al., 2022*). TOPBP1 is composed of nine BRCT domains, which are protein-interacting modules that typically recognize phosphorylated motifs (*Liu et al., 2017*; *Manke et al., 2003*; *Rodriguez et al., 2003*). Through the recognition of phosphoproteins, TOPBP1 is able to assemble multisubunit complexes to promote discrete pathways (*Bigot et al., 2019*; *Blackford et al., 2015*; *Cescutti et al., 2010*; *Delacroix et al., 2007*; *Ellnati et al., 2017*; *Jeon et al., 2019*; *Leimbacher et al., 2019*; *Leung et al., 2011*; *Liu et al., 2017*; *Pereira et al., 2022*; *Pereira et al., 2020*; *Perera et al., 2004*). TOPBP1 interacts with the C-terminal tail of RAD9, a component of the 9-1-1 PCNA-like clamp that is loaded at 5′ recessed junctions adjacent to DSBs (*Parrilla-Castellar et al., 2004*). The 9-1-1-TOPBP1 complex is essential for canonical ATR signaling during prophase I (*Ellnati et al., 2017*; *Jeon et al., 2019*; *Perera et al., 2004*). The interaction of proteins with TOPBP1 may facilitate their phosphorylation by ATR, suggesting a role for TOPBP1 in the control of ATR substrate selection.

As cells progress from zygonema into pachynema, ATR and TOPBP1 localize to the unsynapsed axes of the X and Y chromosomes (*Broering et al., 2014*; *Moens et al., 1999*; *Reini et al., 2004*), leading to phosphorylation of the histone variant H2AX on serine 139 (γH2AX), a major hallmark of MSCI (*Royo et al., 2013*; *Widger et al., 2018*). During establishment of MSCI, a phase-separated structure termed the sex body is formed (*Monesi, 1965*; *Solari, 1974*; *Xu and Qiao, 2021*), allowing the confinement of ATR signaling, DDR factors, and silencing to the X and Y (*Abe et al., 2020*; *Turner, 2007*) as part of a checkpoint that may induce cell death if DDR proteins aberrantly accumulate and remain on autosomes (*Abe et al., 2020*). Recruitment of ATR and TOPBP1 to unsynapsed regions of the XY requires a distinct set of factors compared to their mode of recruitment to autosomal DSB sites mentioned above and involves factors such as BRCA1 and HORMAD (*Shin et al., 2010*; *Turner et al., 2004*). Activated ATR phosphorylates H2AX at the unsynapsed cores of the X and Y chromosomes, a signaling that is propagated to chromatin loops of the X and Y, via a feed-forward process mediated by recruitment of the MDC1 adaptor, which further recruits and mobilizes additional TOPBP1-ATR complexes, therefore spreading ATR signaling to promote the broad chromosome-wide silencing required for MSCI (*Ichijima et al., 2011*). It has also been proposed that the initiation of MSCI accumulates DDR proteins from autosomes to the X and Y chromosomes to prevent excessive DDR signaling at autosomes from activating cellular checkpoints that can stop meiotic progression (*Abe et al., 2020*).

Despite mounting evidence pointing to the importance of ATR and TOPBP1 for MSCI, the precise mechanisms by which they promote sex body formation and XY silencing remain unknown. Moreover, it remains unclear how these two processes are spatiotemporally coordinated and how ATR and TOPBP1 mediate such coordination. Since both proteins are essential for organismal viability (*Brown and Baltimore, 2000*; *de Klein et al., 2000*; *Jeon et al., 2011*; *O'Driscoll, 2009*; *O'Driscoll et al., 2007*; *Yamane et al., 2002*; *Zhou et al., 2013*), conditional knockouts (*Ellnati et al., 2017*; *Royo et al., 2013*; *Widger et al., 2018*) or hypomorphic (*Pacheco et al., 2018*) models have been used to explore their roles during prophase I in spermatocytes. However, given the strong pleiotropic effects in these models, especially in DSB repair, synapsis, MSCI, and sex body formation, it is difficult to dissect the distinct molecular mechanisms involved and untangle direct versus indirect effects. Here we present a separation-of-function mouse mutant that deconvolutes TOPBP1-dependent ATR signaling in male meiosis. We generated mice bearing multiple mutations in BRCT-domain 5 (*Topbp1^{B5/B5}* mice) that are viable and grossly indistinguishable from wild-type littermates; yet, the males are sterile, having reduced testes size, reduced seminiferous tubule cellularity, and a complete loss of sperm. Strikingly, while *Topbp1^{B5/B5}* spermatocytes fail to progress into diplotene, they display largely normal chromosome synapsis, sex body formation, recruitment of DDR proteins to the X and Y, and DNA repair during prophase I, in sharp contrast to previous models of TOPBP1 or ATR impairment (*Ellnati et al., 2017*; *Widger et al., 2018*). Single-cell RNA sequencing data showed that while MSCI is initiated in *Topbp1^{B5/B5}*, the dynamics of silencing progression and reinforcement is defective, which is accompanied by a defect in the localization of the RNA:DNA helicase Senataxin to chromatin loops of the XY chromosomes. We propose that the *Topbp1^{B5/B5}* is a separation of function mutant that allows the untangling of XY silencing from sex body formation and DDR recruitment to the XY, representing a unique model to study the establishment, maintenance, reinforcement, and progression of MSCI.

## Results

### A TOPBP1 mutant separating its role in fertility from organismal viability

*Topbp1* knockout mice exhibit strong defects early in embryonic development, reaching blastocyst stage but not progressing beyond embryonic day (E) 8, with embryos likely dying at the preimplantation stage (*Jeon et al., 2011*). In the context of meiosis, conditional knockout of *Topbp1* in spermatocytes leads to pleiotropic effects, including defects in chromosome synapsis, impaired recruitment of DDR factors to XY chromosomes, defects in condensation of the XY chromosomes, abnormal formation of the sex body, lack of γH2AX spreading to chromatin loops, all of which contribute to a strong MSCI defect as indicated by the complete absence of downstream markers of MSCI, such as USP7, H3K9me3, poly-ubiquitination, and sumoylation (*Ellnati et al., 2017*; *Pereira et al., 2020*). The availability of a separation of function mutant for *Topbp1* is therefore needed to dissect its distinct roles in development, organismal maintenance, and multiple meiotic processes such as DNA repair, silencing, and sex body formation. To generate a separation of function mouse model, we inserted eight charge reversal point mutations in BRCT domain 5 of TOPBP1 (hereafter referred to as *Topbp1^{B5/B5}*) using CRISPR/Cas9 (*Figure 1A*). After validating the point mutations through Sanger sequencing, (*Figure 1—figure supplement 1*), we found that *Topbp1^{B5/B5}* mice were viable, with no difference in body mass when compared to WT littermates (*Figure 1B and C*), and no sensitivity to ionizing radiation (IR) (*Figure 1D*). Strikingly, *Topbp1^{B5/B5}* mice displayed male-specific infertility (*Figure 1E*), with a threefold reduction in testis size (*Figure 1F and G*) and complete lack of spermatozoa (*Figure 1H*). H&E-stained histological testis sections revealed mainly spermatogonia and spermatocytes within the seminiferous epithelium (*Figure 1I*), together with an increased number of TUNEL-positive spermatocytes in seminiferous tubules (*Figure 1J and K*). Cytological evaluation of surface-spread spermatocytes from *Topbp1^{B5/B5}* revealed the presence of meiotic prophase I stages from leptonema to pachynema but a total absence of diplonema-staged spermatocytes (*Figure 1L*). Moreover, the staining of the synaptonemal complex proteins SYCP1 and SYCP3 revealed normal pachytene entry and no gross defects in chromosome synapsis, distinct from previous models of ATR and TOPBP1 conditional depletion (*Ellnati et al., 2017*; *Widger et al., 2018*). Furthermore, unlike reported DDR CKO (conditional knockouts) models such as *Rad1* and *Brca1* (*Abe et al., 2020*; *Broering et al., 2014*;

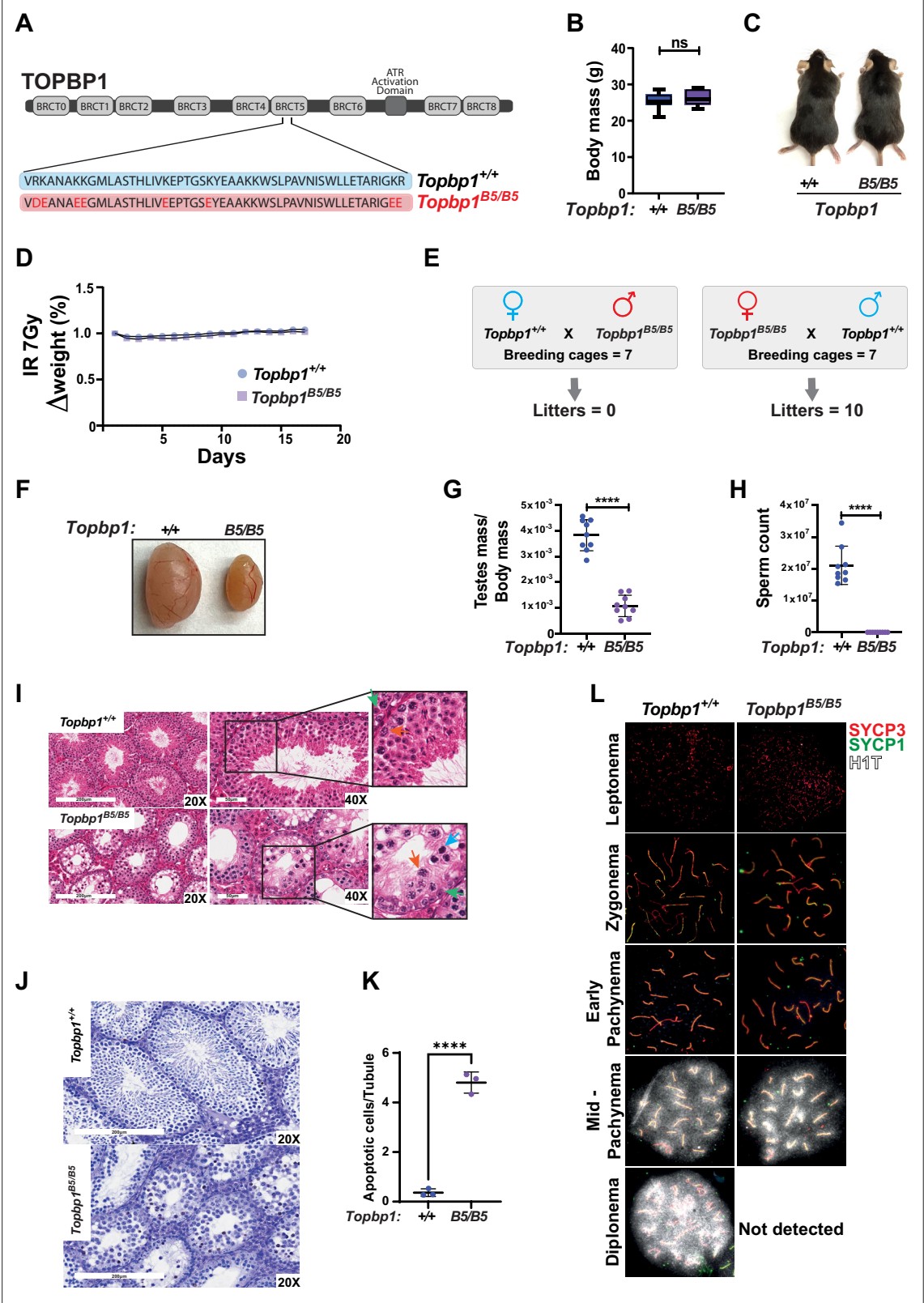

**Figure 1.** A new TOPBP1 mutant separating its role in fertility from organismal viability. (**A**) Schematic showing mutations in the *Topbp1 B5* allele. (**B**) Body mass (*Topbp1^{+/+}* mean = 25.26, SD = 2.38; *Topbp1^{B5/B5}* mean = 26.43, SD = 2.28, n = 9) and (**C**) appearance of *Topbp1^{B5/B5}* and *Topbp1^{+/+}* littermate mice. (**D**) Effect of full body ionizing radiation (IR) (7 Gy) on changes in body mass of *Topbp1^{B5/B5}* and *Topbp1^{+/+}* littermate mice (**E**) Breeding scheme and resulting litters. (**F, G**) Comparison of testes size (*Topbp1^{+/+}* mean = 0.038, SD = 0.006; *Topbp1^{B5/B5}* mean = 0.011, SD = 0.004, n = 9), and (**H**)

*Figure 1 continued on next page*

Figure 1 continued

sperm count, of $Topbp1^{B5/B5}$ and $Topbp1^{+/+}$ littermate mice ($Topbp1^{+/+}$ mean = 2.1 × 10$^7$, SD = 6 × 10$^6$; $Topbp1^{B5/B5}$ mean = 0.0, SD = 0.0, n = 9). (**I**) H&E-stained histological testes sections displaying loss of cellularity in $Topbp1^{B5/B5}$. Green arrow = spermatogonia, red arrow = healthy spermatocyte, blue arrow = dying spermatocyte. (**J**, **K**) TUNEL assay performed on histological testes sections ($Topbp1^{+/+}$ mean = 0.36, SD = 0.15; $Topbp1^{B5/B5}$ mean = 4.80, SD = 0.43, n = 3). (**L**) Meiotic spreads stained for SYCP3, SYCP1, and H1t. ****$p<0.0001$, n = number of mice. p-Values were calculated using unpaired $t$-test.

The online version of this article includes the following figure supplement(s) for figure 1:

**Figure supplement 1.** Genotyping of $Topbp1^{B5/B5}$ mice.

*Pereira et al., 2022*), $Topbp1^{B5/B5}$ pachytene spermatocytes reach mid-pachynema, as demonstrated by the accumulation of signal for H1t (*Inselman et al., 2003*; *Figure 1L*).

## $Topbp1^{B5/B5}$ MEFs display no detectable DDR defects

To assess the possibility of a somatic phenotype, we derived mouse embryonic fibroblasts (MEFs) from $Topbp1^{B5/B5}$ and wild-type littermates at E13.5. Consistent with the observed organismal viability and the lack of IR sensitivity, $Topbp1^{B5/B5}$ MEFs showed no sensitivity to hydroxyurea (replication stress) or phleomycin (DSBs) in a long-term cell survival assay (*Figure 2A–D*, *Figure 2—figure supplement 1*). Genotoxic stress activates the apical kinases ATR, ATM, and DNA-PKcs (*Blackford and Jackson, 2017*; *Falck et al., 2005*; *Maréchal and Zou, 2013*) to trigger a signaling cascade that promotes DNA repair and cell cycle arrest via activation of the downstream checkpoint kinases CHK1 and CHK2 (*Hartwell and Kastan, 1994*; *Lanz et al., 2019*; *Shiloh, 2003*). Similar to wild-type MEFs, $Topbp1^{B5/B5}$ MEFs were able to activate DDR signaling responses when challenged with hydroxyurea and phleomycin as demonstrated by the induction of established damage-induced phosphorylation of CHK1, CHK2, RPA2, and KAP1 (*Figure 2E*). In addition, $Topbp1^{B5/B5}$ MEFs were able to recruit MDC1 to γH2AX-marked DSB foci when subjected to IR (*Figure 2F*) and showed no increased number of micronuclei, a known marker of defective DDRs (*Kwon et al., 2020*), when compared to $Topbp1^{+/+}$ MEFs (*Figure 2—figure supplement 2A and B*).

The BRCT-5 domain of TOPBP1 is known to interact with the DDR factors 53BP1 (*Bigot et al., 2019*; *Cescutti et al., 2010*; *Liu et al., 2017*), MDC1 (*Wang et al., 2011*), and BLM (*Blackford et al., 2015*) through phospho-protein binding modules. To investigate which, if any, of these interactions are disrupted upon mutating the eight BRCT5 residues, we ectopically expressed Flag-TOPBP1-WT or Flag-TOPBP1-B5 in HEK293T cells. We found that binding of Flag-TOPBP1-B5 to BLM and 53BP1 was impaired, as expected (*Figure 2—figure supplement 3A and B*). Moreover, we noticed a twofold reduction in protein levels of Flag-TOPBP1-B5 compared to Flag-TOPBP1-WT, which could be explained by the loss of interaction with BLM (that was proposed to lead to protein stabilization; *Balbo Pogliano et al., 2022*; *Wang et al., 2013*) or by protein misfolding caused by the eight K to E/D mutations. In addition, the reduction in protein levels was detected on MEFs (*Figure 2—figure supplement 4*). In either case, the results presented here show that the TOPBP1-B5 mutant offers a unique model to separate roles of TOPBP1 in male meiosis from the canonical DDR functions of TOPBP1 in somatic cells.

## $Topbp1^{B5/B5}$ spermatocytes display normal markers of canonical ATR signaling, chromosome synapsis, DNA repair, sex body formation, and DDR protein localization at the X and Y

TOPBP1 and ATR play multiple roles in spermatocytes during prophase I. Mice conditionally depleted for TOPBP1 (*Ellnati et al., 2017*) or ATR (*Royo et al., 2013*; *Widger et al., 2018*) display severe defects in chromosome synapsis, DNA DSB repair, sex body formation, and MSCI, as well as impaired recruitment of DDR factors to the XY (*Pereira et al., 2020*). Strikingly, analysis of $Topbp1^{B5/B5}$ spermatocytes via meiotic spreads revealed normal repair of DNA DSBs, with γH2AX staining grossly unchanged at pachynema, being confined only to the XY chromosomes and being excluded from the autosomes (*Figure 3A and B*). RAD51 localized only to the X and Y chromosomes during mid-pachytene (*Figure 3C and D*), indicating normal DSB repair at the autosomes. We were unable to detect chromosome synapsis abnormalities in $Topbp1^{B5/B5}$ spermatocytes as $Topbp1^{B5/B5}$ mutant spermatocytes transit from zygotene into pachytene with normal patterns of HORMAD1 and HORMAD2

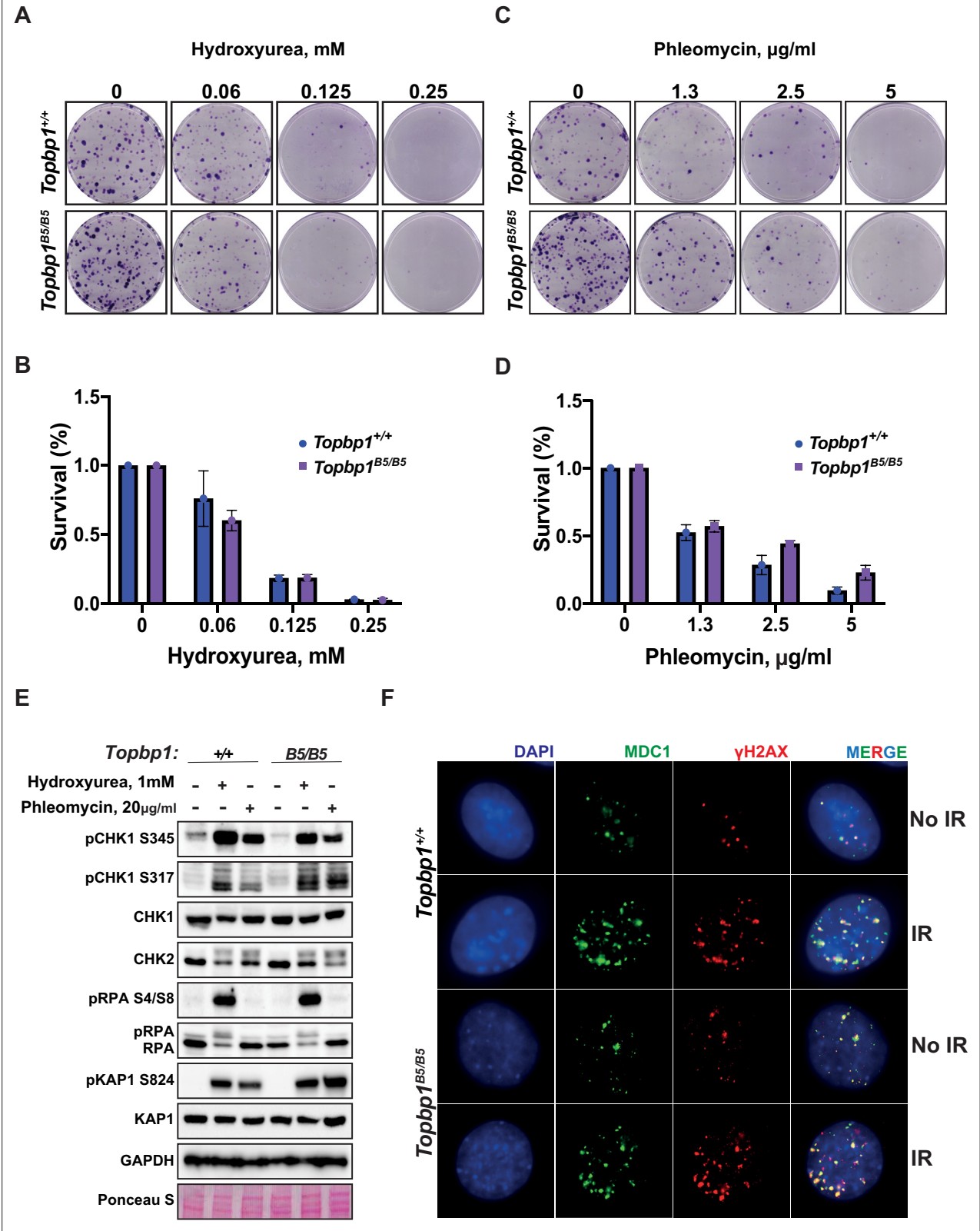

**Figure 2.** *Topbp1^{B5/B5}* mouse embryonic fibroblasts (MEFs) display no detectable DNA damage response (DDR) defects. (**A**) MEFs obtained from littermate mice of indicated genotypes were subjected to clonogenic survival assay in the indicated concentrations of hydroxyurea or (**C**) phleomycin. (**B, D**) Quantification of clonogenic survival assays from (**A**) and (**C**), respectively, performed in biological and experimental triplicates (n = 3), error bars are displayed as standard deviation. (**E**) Western blot for indicated DDR markers in MEFs obtained from *Topbp1^{B5/B5}* and *Topbp1^{+/+}* littermate mice.

*Figure 2 continued on next page*

*Figure 2 continued*

The data from MEFs were performed using littermate pairs and validated using a second pair of *Topbp1^{B5/B5}* and *Topbp1^{+/+}* littermate mice.
(**F**) Immunofluorescence of MDC1 and phosphoH2AX_S139-stained nuclei from *Topbp1^{B5/B5}* and *Topbp1^{+/+}* MEFs treated with ionizing radiation (IR) (7 Gy).

The online version of this article includes the following source data and figure supplement(s) for figure 2:

**Source data 1.** Original file for the western blot analysis in *Figure 2E* (anti-CHK1, anti-CHK2, and anti-GAPDH).

**Figure supplement 1.** Survival assays of *Topbp1^{B5/B5}* mouse embryonic fibroblasts (MEFs).

**Figure supplement 2.** Absence of an increased number of micronuclei in *Topbp1^{B5/B5}* mouse embryonic fibroblasts (MEFs).

**Figure supplement 3.** TOPBP1-B5 shows disruption in binding 53BP1 and BLM.

**Figure supplement 3—source data 1.**

**Figure supplement 4.** *Topbp1^{B5/B5}* mouse embryonic fibroblasts (MEFs) show lower TOPBP1 protein abundance than *Topbp1^{+/+}* MEFs.

**Figure supplement 4—source data 1.** Original file for the western blot analysis in *Figure 2—figure supplement 4* (anti-TOPBP1 and anti-TUBULIN).

**Figure supplement 4—source data 2.** PDF containing *Figure 2—figure supplement 4* and original scans of the relevant western blot analysis (anti-TOPBP1 and anti-TUBULIN) with highlighted bands and labels.

localization (*Figure 3—figure supplement 1A and B*) and with SYCP1 overlapping with SYCP3 on chromosome cores from all autosomal chromosomes during pachynema (*Figures 1L and 3E*). Since ATR orthologs regulate crossing over in budding yeast (*Subramanian and Hochwagen, 2014*) and in *Drosophila melanogaster* (*Carpenter, 1979*), we also investigated the localization of factors involved in regulating DNA crossovers, including MLH1 and MLH3. In *Topbp1^{B5/B5}* spermatocytes, while the number of MLH1 foci were significantly increased, the number of MLH3 foci did not differ significantly (*Figure 3—figure supplement 2A and B*). Due to the lack of diplotene cells, and any other stage after pachynema, we were not able to test whether *Topbp1^{B5/B5}* display defects in crossing over.

The sex body appeared normal in its shape and was normally formed in *Topbp1^{B5/B5}* spermatocytes (*Figure 3A*, *Figure 3—figure supplements 3 and 4A*), exhibiting only subtle defects/delays in the spreading of the γH2AX signal on the PAR and pericentromeric regions (*Figure 3—figure supplement 4B–G*). Although the defect was subtle in its severity, it accounted for 56% of all γH2AX-stained mid-pachytene cells. Importantly, TOPBP1 was normally localized to X and Y chromosomes during prophase I (*Figure 3—figure supplement 5*, *Figure 3F–H*). Similarly, localization of the TOPBP1 inter-actors ATR, BRCA1, MDC1, and 53BP1 in *Topbp1^{B5/B5}* spermatocytes spreads was also indistinguish-able from that observed in *Topbp1^{+/+}* spermatocytes (*Figure 4A–F*, *Figure 4—figure supplement 1*). Markers of ATR signaling were also mostly normal, as measured by its canonical targets pMDC1 T4 (*Figure 3—figure supplement 1C and D*), pCHK1 S345 (*Figure 3—figure supplement 1E and F*), and pHORMAD2 (*Figure 3—figure supplement 1G and H*). Notably, we did observe that phosphorylation of CHK1 on S317 was significantly decreased in *Topbp1^{B5/B5}* when compared to *Topbp1^{+/+}* spermatocytes (*Figure 4G and H*). However, since *Chek1* CKO spermatocytes complete prophase I and differentiate into spermatozoa, with only minor defects such as a delay in the removal of γH2AX from autosomes (*Abe et al., 2018*), the observed defect in CHK1 S317 phosphorylation is unlikely to be the cause of the infertility observed in *Topbp1* B5 mutants. Overall, as summarized in *Figure 4I*, *Topbp1^{B5/B5}* spermatocytes appear to progress normally through early stages of prophase I up until the end of pachynema as demonstrated by largely normal localization of markers for DNA repair, chromosome synapsis, and ATR signaling. These findings are surprising because the lack of sex body formation, synapsis defects, or unrepaired DSBs, which are the expected causes of the drastic loss of diplotene cells and the lack of spermatozoa, were not observed.

## Defective phosphorylation and XY localization of the RNA:DNA helicase SETX in *Topbp1^{B5/B5}* spermatocytes

With the exception of CHK1 phosphorylation at serine 317, the analysis of other canonical markers of ATR signaling on meiotic prophase I spreads did not reveal obvious defects in their distribution or intensity at the XY body (*Figure 4G and H*, *Figure 3—figure supplement 1C–H*). Since altered CHK1 regulation is unlikely to be the cause of the drastic defect in meiotic progression observed in *Topbp1^{B5/B5}* males, we investigated whether other branches of ATR signaling were altered in *Topbp1^{B5/B5}* spermatocytes by performing unbiased quantitative phosphoproteomics based on TMT (Tandem

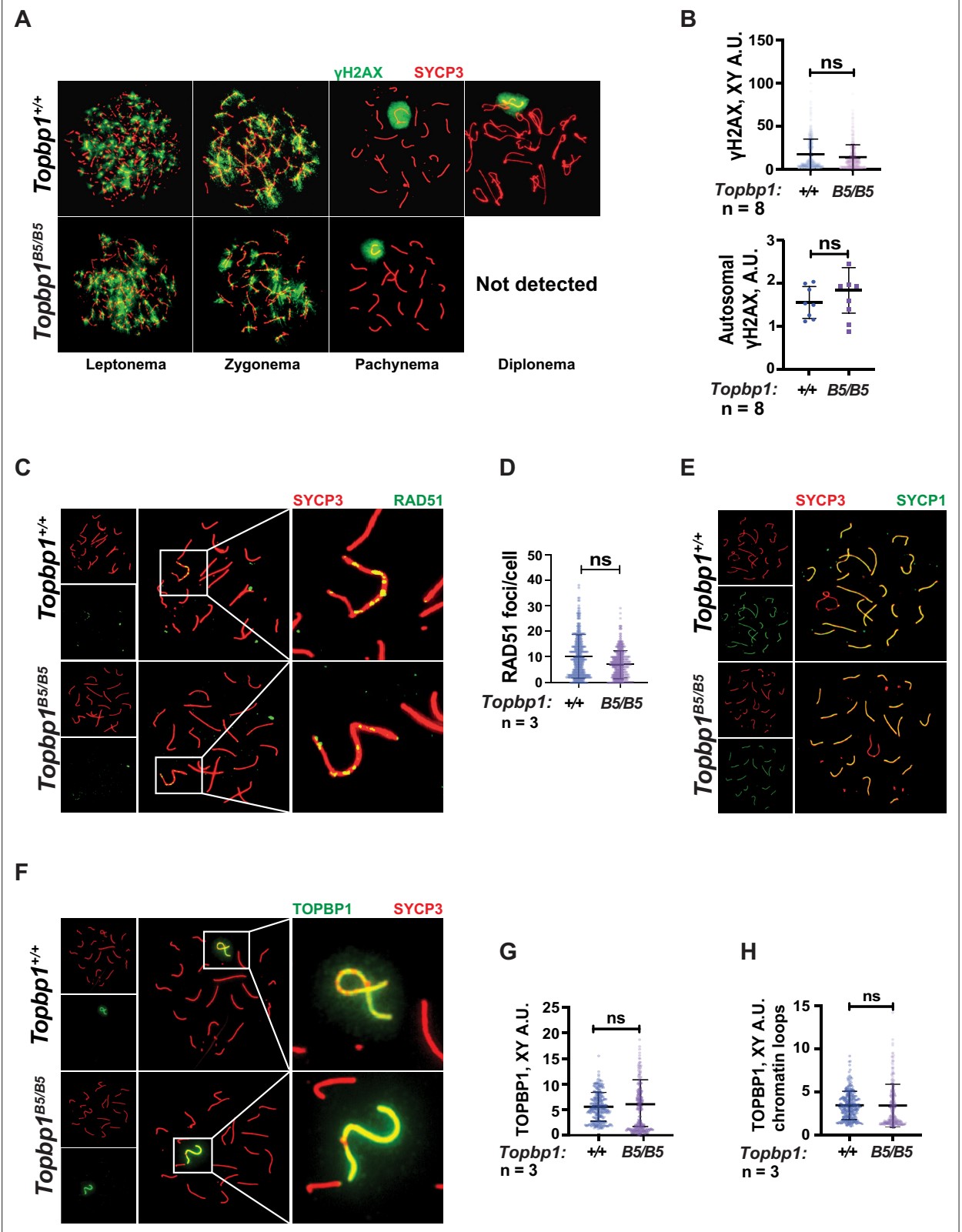

**Figure 3.** Markers of DNA repair, synapsis, sex body formation, and TOPBP1 localization are mostly normal in *Topbp1^{B5/B5}* spermatocytes. (**A**) Meiotic spreads showing *Topbp1^{+/+}* and *Topbp1^{B5/B5}* spermatocytes stained with SYCP3 and γH2AX, prepared as described in 'Materials and methods'. (**B**) Quantification of γH2AX-stained pachytene spreads, upper graph XY body (each dot represents one pachytene cell measured; *Topbp1^{+/+}* number of cells = 386, number of mice = 8; *Topbp1^{B5/B5}* number of cells = 410, number of mice = 8; p-value=0.3063), bottom graph autosomal chromosomes

*Figure 3 continued on next page*

*Figure 3 continued*

(each dot represents the average of signal from all autosomes in each mouse, *Topbp1*$^{+/+}$ number of cells = 160, number of mice = 8; *Topbp1*$^{B5/B5}$ number of cells = 161, number of mice = 8; p-value=0.5081). (**C**) Meiotic spreads showing *Topbp1*$^{+/+}$ and *Topbp1*$^{B5/B5}$ pachytene spermatocytes stained with SYCP3 and RAD51. (**D**) Quantification of RAD51 foci/cell of mid-pachytene meiotic spreads (each dot represents one pachytene cell measured; *Topbp1*$^{+/+}$ number of cells = 149, number of mice = 3; *Topbp1*$^{B5/B5}$ number of cells = 183, number of mice = 3; p-value=0.2174). (**E**) Meiotic spreads showing *Topbp1*$^{+/+}$ and *Topbp1*$^{B5/B5}$ pachytene spermatocytes stained with SYCP3 and SYCP1. (**F**) Meiotic spreads showing *Topbp1*$^{+/+}$ and *Topbp1*$^{B5/B5}$ pachytene spermatocytes stained with SYCP3 and TOPBP1. (**G**) Quantification of TOPBP1 on X and Y chromosome cores from (**F**) (each dot represents one pachytene cell measured; *Topbp1*$^{+/+}$ number of cells = 246, number of mice = 3; *Topbp1*$^{B5/B5}$ number of cells = 233, number of mice = 3; p-value=0.8546). (**H**) Quantification of TOPBP1 on X and Y chromatin loops from (**F**) (each dot represents one pachytene cell measured; *Topbp1*$^{+/+}$ number of cells = 246, number of mice = 3; *Topbp1*$^{B5/B5}$ number of cells = 233, number of mice = 3; p-value=0.6755). n = number of mice. p-Values were calculated using a linear mixed effect model (see 'Materials and methods' for details).

The online version of this article includes the following figure supplement(s) for figure 3:

**Figure supplement 1.** *Topbp1*$^{B5/B5}$ mid-pachytene spermatocytes show normal HORMAD1, HORMAD2, pHORMAD2, and pCHK1_S345 localization and intensities.

**Figure supplement 2.** *Topbp1*$^{B5/B5}$ mid-pachytene spermatocytes show an increased number of MLH1 foci/cell but no difference in the number of MHL3 foci/cell.

**Figure supplement 3.** *Topbp1*$^{B5/B5}$ mid-pachytene spermatocytes form a normal sex body.

**Figure supplement 4.** *Topbp1*$^{B5/B5}$ mid-pachytene spermatocytes show subtle defects/delays in the spreading of γH2AX on XY chromosomes.

**Figure supplement 5.** TOPBP1 is normally localized in all stages of prophase I in *Topbp1*$^{B5/B5}$.

Mass Tag; *Thompson et al., 2003*) labeling of whole testes. Following a similar approach previously used by our group (*Sims et al., 2022*), we analyzed the phosphoproteome of *Topbp1*$^{B5/B5}$ and *Topbp1*$^{+/+}$ testes, and then compared the results with our previously reported dataset comparing the phosphoproteome of testes from mice treated with vehicle or the ATR inhibitor AZ20 (*Sims et al., 2022*; *Figure 5A*). The resulting plot revealed that ATR-dependent signaling is not drastically impaired in *Topbp1*$^{B5/B5}$ testes, as opposed to the marked impairment of ATR signaling previously observed in testes of *Rad1* CKO mice (*Sims et al., 2022*). This finding is in agreement with the results from meiotic spreads of *Topbp1*$^{B5/B5}$ showing no detectable defects in canonical markers of ATR signaling described above. Nonetheless, our phosphoproteomic analysis did reveal phosphorylation sites mildly disrupted in *Topbp1*$^{B5/B5}$ testes compared to *Topbp1*$^{+/+}$ testes. In particular, we noticed that several phosphorylation sites in the RNA:DNA helicase, Senataxin (SETX) (*Cohen et al., 2018*), including a phosphorylation in the preferred motif for ATR phosphorylation (S/T-Q), were downregulated in *Topbp1*$^{B5/B5}$ mice (*Figure 5B*). Interestingly, SETX was previously associated with meiotic ATR functions and found to be indispensable for MSCI (*Becherel et al., 2013*; *Yeo et al., 2015*). Moreover, we recently reported that mice treated with the ATR inhibitor AZ20 or lacking *Rad1* display reduced phosphorylation of SETX at S/T-Q site and SETX mislocalization during pachynema (*Sims et al., 2022*). Based on our findings, we propose that TOPBP1 regulates SETX distribution and/or function during meiosis, and that defects in meiotic progression and fertility observed in *Topbp1*$^{B5/B5}$ mice might be associated with SETX dysfunction. Consistent with this possibility, we found that pachytene spermatocytes from *Topbp1*$^{B5/B5}$ display significantly decreased spreading of SETX to XY chromatin loops while still displayed SETX at the unsynapsed axes of the X and Y chromosomes (*Figure 5C–F*). Overall, these findings reveal that *Topbp1*$^{B5/B5}$ spermatocytes display largely normal progression through mid-pachynema, as demonstrated by normal distribution of a range of markers of meiotic progression, including canonical markers of ATR signaling. However, in-depth phosphoproteomic and imaging analyses identify specific defects in the regulation of SETX, a target of ATR signaling during prophase I (*Pereira et al., 2022*; *Sims et al., 2022*) and a key factor required for MSCI (*Becherel et al., 2013*; *Yeo et al., 2015*).

## *Topbp1*$^{B5/B5}$ spermatocytes initiate MSCI but fail to promote full XY silencing

During mid-pachynema, spermatocytes that fail to properly silence the X and Y chromosomes arrest and trigger apoptosis-induced cell death (*Abe et al., 2020*; *Ichijima et al., 2012*; *Turner, 2015*; *Turner, 2007*). In mice, the MSCI process initiates in leptonema (*Lau et al., 2020*), and during early pachynema key events occur at the XY chromosomes, such as exclusion of RNA polymerase 2 (RNA

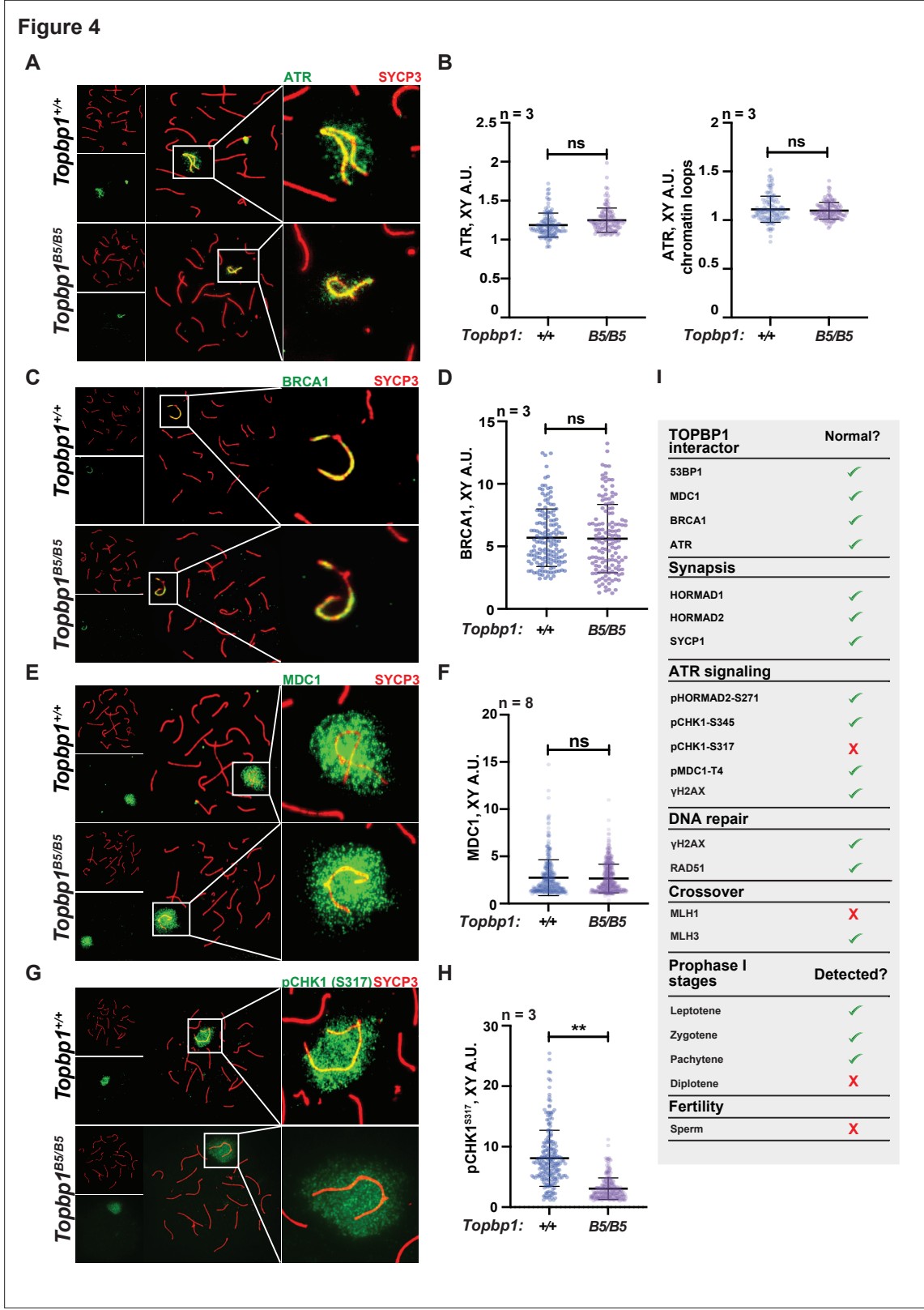

**Figure 4.** *Topbp1^{B5/B5}* spermatocytes display normal localization of ATR, BRCA1, and MDC1. (**A**) Meiotic spreads showing *Topbp1^{+/+}* and *Topbp1^{B5/B5}* pachytene spermatocytes stained with SYCP3 and ATR. (**B**) Quantification of ATR in pachytene spreads from (**A**). Left: ATR on X and Y chromosome cores (each dot represents one pachytene cell measured; *Topbp1^{+/+}* number of cells = 127, number of mice = 3; *Topbp1^{B5/B5}* number of cells = 127, number of mice = 3; p-value=0.4068). Right: ATR on X and Y chromatin loops (each dot represents one pachytene cell measured; *Topbp1^{+/+}*

*Figure 4 continued on next page*

*Figure 4 continued*

number of cells = 127, number of mice = 3; *Topbp1^{B5/B5}* number of cells = 127, number of mice = 3; p-value=0.9396). (**C**) Meiotic spreads showing *Topbp1^{+/+}* and *Topbp1^{B5/B5}* pachytene spermatocytes stained with SYCP3 and BRCA1. (**D**) Quantification of BRCA1 in pachytene spreads from (**C**) (each dot represents one pachytene cell measured; *Topbp1^{+/+}* number of cells = 152, number of mice = 3; *Topbp1^{B5/B5}* number of cells = 140, number of mice = 3; p-value=0.6509). (**E**) Meiotic spreads showing *Topbp1^{+/+}* and *Topbp1^{B5/B5}* pachytene spermatocytes stained with SYCP3 and MDC1. (**F**) Quantification of MDC1 in pachytene spreads from (**E**) (each dot represents one pachytene cell measured; *Topbp1^{+/+}* number of cells = 696, number of mice = 8; *Topbp1^{B5/B5}* number of cells = 988, number of mice = 8; p-value=0.3603). (**G**) Meiotic spreads showing *Topbp1^{+/+}* and *Topbp1^{B5/B5}* pachytene spermatocytes stained with SYCP3 and pCHK1-S317. (**H**) Quantification of pCHK1-S317 in pachytene spreads from (**G**) (each dot represents one pachytene cell measured; *Topbp1^{+/+}* number of cells = 223, number of mice = 3; *Topbp1^{B5/B5}* number of cells = 254, number of mice = 3; \*\*p-value=0.0023). p-Values were calculated using a linear mixed effect model (see 'Materials and methods' for details). (**I**) Table summarizing the normal or disrupted ATR and TOPBP1-dependent events during male fertility accessed in *Topbp1^{B5/B5}*. n = number of mice.

The online version of this article includes the following figure supplement(s) for figure 4:

**Figure supplement 1.** 53BP1 localization in *Topbp1^{B5/B5}* mid-pachytene spermatocytes.

Pol2), recruitment of DDR proteins and chromatin remodelers, and establishment of heterochromatin marks (***Abe et al., 2020***; ***Khalil et al., 2004***) to maintain the active silencing of the X and Y chromosomes (***Abe et al., 2022***). This process leads to the formation of a membrane-less phase separated structure termed the sex body (***Alavattam et al., 2021***; ***Xu and Qiao, 2021***). Spermatocytes from *Topbp1^{B5/B5}* males formed a sex body of grossly normal appearance (***Figure 3—figure supplement 3***), with undisrupted patterns of the sex body markers CHD4, SUMO, and USP7 (***Figure 6—figure supplement 1***). *Topbp1^{B5/B5}* pachytene spermatocytes also display normal patterns of a range of chromatin marks, including H3K9ac, H3K9me3, H3K27ac, H3K36me3, H3K4me3, H4K16ac, H4ac, H2AK116ub, H3K4me1, as well as proper exclusion of RNA Pol2 from the sex body (***Figure 6—figure supplements 2–6***). Collectively the evidence presented herein shows that *Topbp1^{B5/B5}* mutants are able to form grossly normal XY bodies, with proper localization of over 28 markers. The severe loss of diplotene cells and the reduction of SETX at the chromatin loops of the X and Y lead us to speculate that *Topbp1^{B5/B5}* spermatocytes may still have a defective MSCI. To investigate potential MSCI defects, we employed single-cell RNA sequencing (scRNAseq) in germ cells (***Figure 6A***) using a similar approach recently used to follow spermatogenesis progression and evaluate MSCI in mammals (***Grive et al., 2019***; ***Jung et al., 2019***; ***Lau et al., 2020***). Using the 10X platform, we performed scRNAseq on a germ cell-enriched population of cells extracted from adult *Topbp1^{+/+}* and *Topbp1^{B5/B5}* testes (***Figure 6A***). The data were analyzed as previously described (***Grive et al., 2019***; ***Lau et al., 2020***) using signature genes as markers of different stages in spermatogenesis such as *Sal4* and *Dmrt1* for spermatogonia, *Dazl* for early spermatocytes, *Id4*, *Sycp3* and *Shcbp1l* for late spermatocytes, *Acrv1* for round spermatids, and *Oaz3* and *Prm2* for elongated spermatids (***Figure 6—figure supplements 7 and 8***). *Col1a2*, *Acta2*, *Vcam1*, *Lnsl3*, *laptm5*, *Hbb-bt*, *Ptgds,* and *Wt1* were used as markers of somatic cells (***Figure 6—figure supplement 9***). Analysis of the testicular transcriptome of *Topbp1^{+/+}* males revealed 47 sub-clusters of cells covering spermatogonia, spermatocytes, spermatids, and somatic cells (***Figure 6B–D***, ***Figure 6—figure supplements 10 and 11***). In sharp contrast, analysis of *Topbp1^{B5/B5}* germ cell population revealed only somatic, spermatogonia, and the initial populations of spermatocytes (***Figure 6D***). This result corroborates the H&E-stained testes histological sections (***Figure 1I***) and meiotic spreads (***Figure 1L***). Importantly, Pearson correlation values from all RNA reads between cell groups separated by cluster and genotype, demonstrated the similarity of spermatocytes from *Topbp1^{+/+}* and *Topbp1^{B5/B5}* (***Figure 6—figure supplement 12***). As shown in ***Figure 6E***, we were able to monitor the dynamics of X chromosome silencing in the early-stage spermatocytes and compare the results between RNA from *Topbp1^{+/+}* and *Topbp1^{B5/B5}* males. Strikingly, *Topbp1^{B5/B5}* early-stage spermatocytes could initiate MSCI and promote robust, albeit incomplete, X chromosome silencing. Both X and Y chromosomes showed increased gene expression levels in the last spermatocyte stage captured in *Topbp1^{B5/B5}* males when compared to RNA from wild-type males (***Figure 6E***, ***Figure 6—figure supplement 13***). Moreover, although certain X-genes from *Topbp1^{B5/B5}* pachytene cells consistently demonstrated defects in silencing by being expressed in numerous cells, other genes were only expressed in a small number of cells. This highlights the non-uniformity of the MSCI defect in all pachytene cells (***Figure 6—figure supplement 13***). Notably, while we detected only a minor elevation in the levels of X-linked gene expression in *Topbp1^{B5/B5}* spermatogonia when compared to *Topbp1^{+/+}*, the expression of X-linked genes at spermatocyte 3 stage was drastically

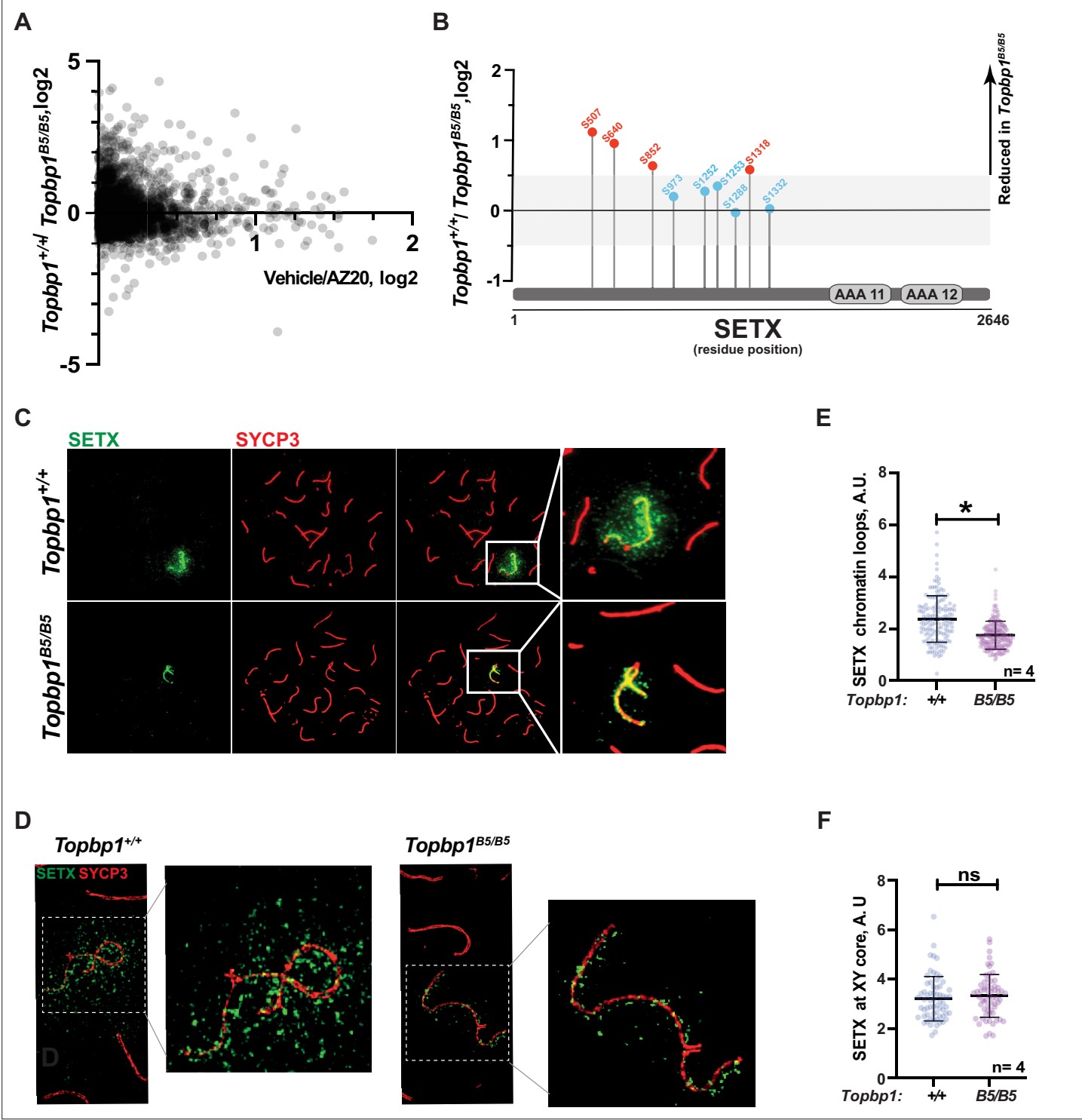

**Figure 5.** Defective Senataxin (SETX) phosphorylation and localization in *Topbp1^{B5/B5}* spermatocytes. (**A**) Scatter plot of phosphoproteomic datasets corresponding to *Topbp1^{+/+}*/*Topbp1^{B5/B}* (Y axis) and *Topbp1^{+/+}*(vehicle)/*Topbp1^{+/+}*(AZ20) (X axis) from whole testes of mice. (**B**) SETX phosphopeptides identified in the *Topbp1^{+/+}*/*Topbp1^{B5/B}* phosphoproteomic experiment shown in (**A**). Red: reduced in *Topbp1^{B5/B5}* mutant; blue: unchanged. (**C**) Meiotic spreads showing pachytene spermatocytes from *Topbp1^{+/+}* and *Topbp1^{B5/B5}* mice stained with SYCP3 and SETX in regular immunofluorescence. (**D**) 3D-SIM analysis of meiotic spreads described in (**C**). (**E**) Quantification of SETX on X and Y chromatin loops in pachytene spreads from (**C**) (each dot represents one pachytene cell measured; *Topbp1^{+/+}* number of cells = 152, number of mice = 4; *Topbp1^{B5/B5}* number of cells = 174, number of mice = 4; *p-value=0.0452). (**F**) Quantification of SETX on X and Y chromosome cores in pachytene spreads from (**C**) (each dot represents one pachytene cell measured; *Topbp1^{+/+}* number of cells = 152, number of mice = 4; *Topbp1^{B5/B5}* number of cells = 174, number of mice = 4; p-value=0.5987). p-Values were calculated using a linear mixed effect model (see 'Materials and methods' for details).

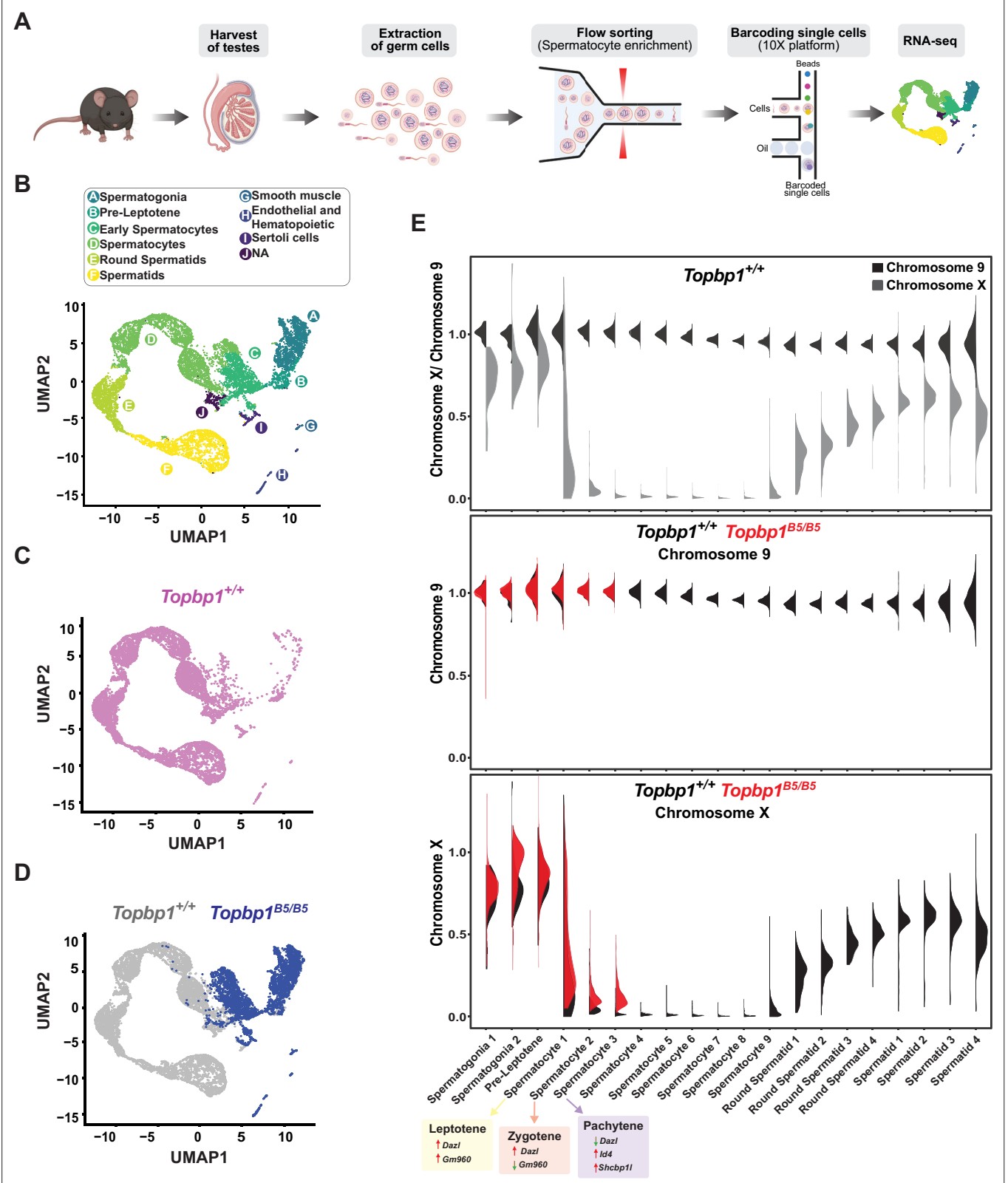

**Figure 6.** Single-cell RNAseq reveals that *Topbp1^B5/B5* spermatocytes initiate meiotic sex chromosome inactivation (MSCI) but fail to promote full silencing. (**A**) scRNAseq workflow for isolation and purification of single cells for RNAseq. (**B**) Uniform Manifold Approximation and Projection (UMAP) analysis of sub-clusters captured in the scRNAseq, representing all cells captured for both *Topbp1^+/+* and *Topbp1^B5/B5*. (**C**) UMAP analysis of sub-clusters captured in the scRNAseq of *Topbp1^+/+*. (**D**) UMAP analysis of sub-clusters captured in the scRNAseq of *Topbp1^+/+* (gray) and *Topbp1^B5/B5* (blue). (**E**) Violin

*Figure 6 continued on next page*

*Figure 6 continued*

plots displaying the ratio of the average expression of X chromosome genes by the average expression of chromosome 9 genes at different stages of spermatogenesis for *Topbp1⁺/⁺* and *Topbp1^B5/B5* cells. The level of X-genes expression in spermatocyte 3 is significantly higher in *Topbp1^B5/B5* cells when compared to *Topbp1⁺/⁺* cells, with a p-value of 1.5e-178 using a two-sided Wilcoxon rank-sum test.

The online version of this article includes the following figure supplement(s) for figure 6:

**Figure supplement 1.** *Topbp1^B5/B5* pachytene spermatocytes do not exhibit any difference in CHD4, Sumo2-3, and USP7 intensities and localization compared to *Topbp1⁺/⁺*.

**Figure supplement 2.** *Topbp1^B5/B5* pachytene spermatocytes show no difference in H3K9ac or K3K9me3 staining compared to *Topbp1⁺/⁺*.

**Figure supplement 3.** *Topbp1^B5/B5* pachytene spermatocytes display no difference in H3K27ac, H3K36me3, or K3K4me3 staining compared to *Topbp1⁺/⁺*.

**Figure supplement 4.** *Topbp1^B5/B5* pachytene spermatocytes display no difference in H4K16ac, H4ac, or H2AK166ub staining compared to *Topbp1⁺/⁺*.

**Figure supplement 5.** From early pachytene to beginning of late pachytene, *Topbp1^B5/B5* spermatocytes exhibit the same localization pattern of H3K4me1 compared to *Topbp1⁺/⁺*.

**Figure supplement 6.** *Topbp1^B5/B5* pachytene spermatocytes exhibit no differences in the intensity or localization patterns of RNApol2, pRNApol2_Ser2, or pRNApol2_ser5 compared to *Topbp1⁺/⁺*.

**Figure supplement 7.** Genes used as markers to define the stages of the germ cells in the scRNAseq analysis.

**Figure supplement 8.** Genes used as markers to track spermatogenesis progression and form the 47 sub-cluster for all cells captured in the 10X platform in the scRNAseq analysis.

**Figure supplement 9.** Genes used as markers of somatic cells captured in the scRNAseq analysis.

**Figure supplement 10.** Analysis of sub-clusters of all cells captured in the 10X platform.

**Figure supplement 11.** Analysis of the main 24 sub-clusters captured in the germ cells-enriched scRNAseq analysis.

**Figure supplement 12.** Spermatocytes from *Topbp1⁺/⁺* and *Topbp1^B5/B5* show correlation greater than 0.9.

**Figure supplement 13.** *Topbp1^B5/B5* pachytene spermatocytes show increased expression of X-linked genes.

**Figure supplement 14.** *Topbp1^B5/B5* pachytene spermatocytes show increased expression of XY 'killer genes' and other X and Y genes typically used to illustrate meiotic sex chromosome inactivation (MSCI) defects.

**Figure supplement 15.** *Topbp1^B5/B5* pachytene spermatocytes show increased expression of XY-linked genes.

**Figure supplement 16.** Quality control of the scRNAseq data.

---

higher in *Topbp1^B5/B5* males when compared to *Topbp1⁺/⁺* males. To improve the accuracy of the downstream analysis, the expression levels of X-linked genes were normalized by their respective expression level at the pre-leptotene stage.

MSCI is a dynamic process that involves the accumulation of DDR factors at the X and Y chromosomes as cells enter pachynema (*Abe et al., 2022*; *Abe et al., 2020*), as well as the inactivation of specific X and Y genes that lead to cell death if expressed at this stage (the so-called 'killer genes') (*Royo et al., 2010*; *Vernet et al., 2016*). Similar to previous reports based on mutants or treatments that impaired MSCI (*Abe et al., 2022*; *Ellnati et al., 2017*; *Hirota et al., 2018*; *Modzelewski et al., 2012*; *Pereira et al., 2022*; *Royo et al., 2013*; *Widger et al., 2018*), transcriptomics profiles from *Topbp1^B5/B5* testicular cells showed an increased number of stage 3 spermatocytes (SP3 – pachytene) expressing the spermatocyte-toxic genes *Zfy1* and *Zfy2* compared to *Topbp1⁺/⁺* (*Figure 6—figure supplements 14 and 15*). Other genes typically used to illustrate MSCI defects, such as *Kdm6a*, *lamp2*, *Zfx*, *Uba1y*, and *Rhox13*, were also expressed in a higher number of SP3 cells in *Topbp1^B5/B5* (*Figure 6—figure supplements 14 and 15*). In the case of *Scml2*, *Topbp1^B5/B5* cells not only displayed an increased number of SP3 (pachytene) cells expressing it but also displayed an increase in expression levels in these cells compared to *Topbp1⁺/⁺* testes (*Figure 6—figure supplements 14 and 15*). In summary, various Y and X genes that had previously been shown to be expressed in other MSCI-defective mutants were found de-repressed in *Topbp1^B5/B5* spermatocytes.

Detailed analysis of the scRNAseq data for the X-linked genes monitored during the early spermatocyte stages revealed important differences in the silencing dynamics of these X genes between *Topbp1^B5/B5* and *Topbp1⁺/⁺* spermatocytes. Out of the roughly 700 genes present on the X chromosome, 233 had reads detected from pre-leptotene to spermatocyte 3 clusters for both *Topbp1⁺/⁺* and *Topbp1^B5/B5* cells, and therefore were used during the downstream analysis. As shown in *Figure 7A*, we clustered the X-linked genes into three distinct categories based on the change of RNA level between the stages: reduced, unaltered, or increased silencing (*Figure 7A*). SP1 (spermatocyte 1) was

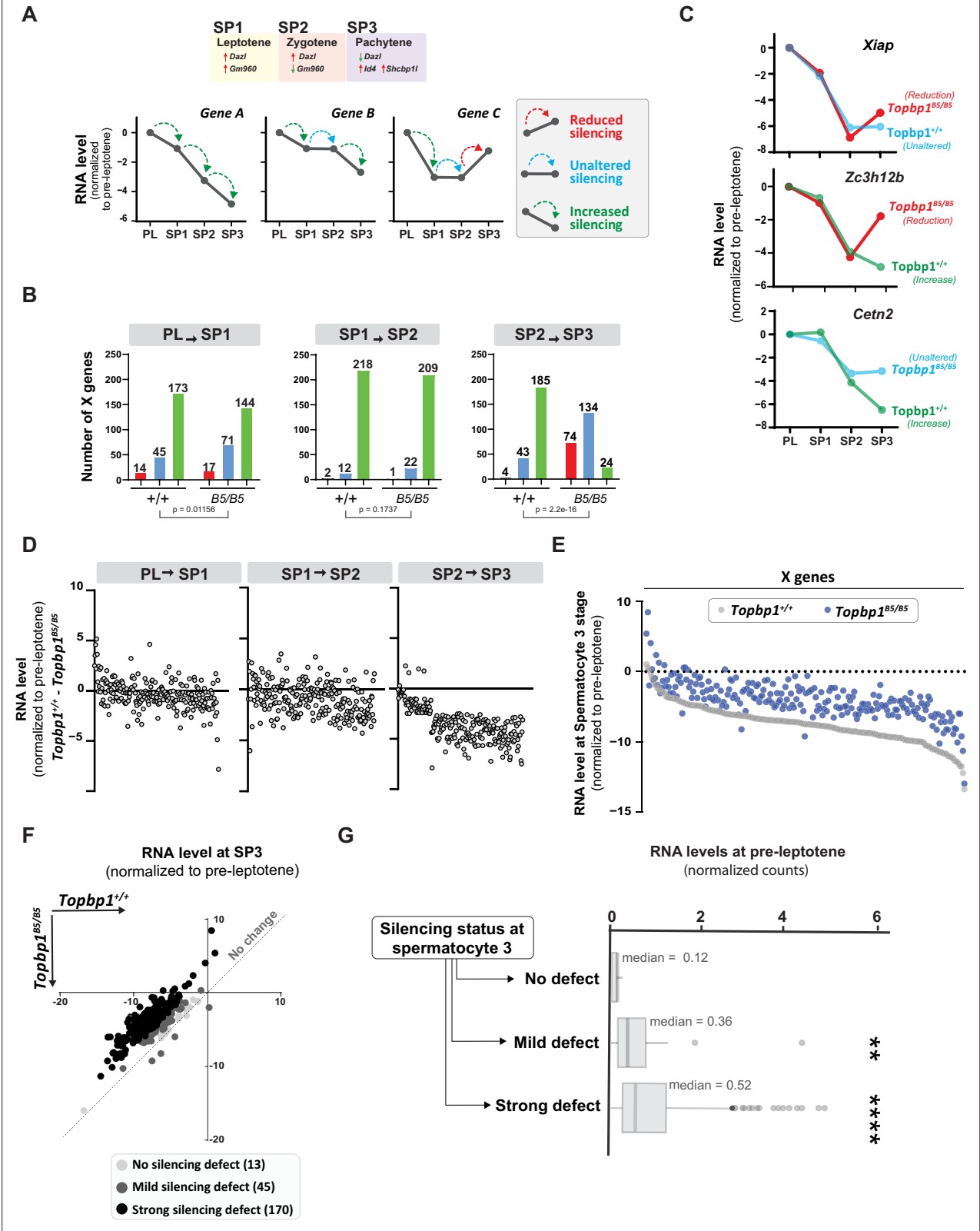

**Figure 7.** TOPBP1 regulates silencing dynamics of X genes at the spermatocyte 3 stage. (**A**) Illustration of the gene markers used to define spermatocyte 1 as leptotene, spermatocyte 2 as zygotene, and spermatocyte 3 as pachytene; hypothetical examples illustrating the categorization of transitions in silencing dynamics between the stages of pre-leptotene (PL), spermatocyte 1 (SP1), spermatocyte 2 (SP2), and spermatocyte 3 (SP3). (**B**) Number of genes in each of the categories described in (**A**), during the different stage transitions and respective p-values above each graph (the

*Figure 7 continued on next page*

*Figure 7 continued*

p-values were calculated using the Fisher's exact test). (**C**) Examples of genes with altered silencing dynamics in the *Topbp1^B5/B5^*, red = reduced silencing, blue = unaltered silencing and green = increased silencing (**D**) Scatter plot showing the difference in RNA level between *Topbp1^+/+^* and *Topbp1^B5/B5^* for each of the indicated stage transitions. (**E**) Scatter plot showing expression level of X-chromosome genes, normalized to pre-leptotene levels, in *Topbp1^+/+^* (gray) and *Topbp1^B5/B5^* (blue) at SP3. (**F**) Graph plotting expression levels of X-chromosome genes, normalized to pre-leptotene levels, in *Topbp1^+/+^* (Y axis) and *Topbp1^B5/B5^* (X axis) and split in three categories based on the severity of silencing defect. (**G**) Box plot showing PL expression levels of X-chromosome genes in each of the categories of silencing defect severity shown in (**F**).

The online version of this article includes the following figure supplement(s) for figure 7:

**Figure supplement 1.** Analysis of the gene marker used to define the zygotene stage.

**Figure supplement 2.** Expression levels of genes with no defects, mild defects, and strong defects in *Topbp1^B5/B5^* and *Topbp1^+/+^*.

defined as leptonema using the gene marker *Gm960* (***Chen et al., 2018***; ***Figure 7—figure supplement 1***), and SP2 (spermatocyte 2) as zygonema due to its profile of low expression of *Gm960* and high expression of *Dazl*. SP3 (spermatocyte 3) was defined as pachynema due to its lower expression of *Dazl* and increased expression of *Id4* and *Shcbp1l*. When comparing the distribution of genes in these clusters between *Topbp1^B5/B5^* and *Topbp1^+/+^* spermatocytes (RNA level normalized to pre-leptotene stage) (***Figure 7B***), we observed no major differences in the pre-leptotene to spermatocyte 1 (PL to SP1) and in the spermatocyte 1 to spermatocyte 2 (SP1 to SP2) transitions. In contrast, when comparing the spermatocyte 2 to spermatocyte 3 (SP2 to SP3) transition between *Topbp1^B5/B5^* and *Topbp1^+/+^* spermatocytes we noticed a major difference in the distribution of the clusters, with 74 genes in *Topbp1^B5/B5^* versus 4 in *Topbp1^+/+^* exhibiting reduced silencing (***Figure 7B***). Moreover, while 134 genes showed unaltered silencing, and only 24 increased silencing in *Topbp1^B5/B5^* spermatocytes during the SP2 to SP3 transition, 43 genes showed unaltered silencing and 185 increased silencing in *Topbp1^+/+^* (***Figure 7B***). *Xiap*, *Zc3h12b*, and *Cetn2* are examples of X-linked genes displaying altered silencing behaviors in *Topbp1^B5/B5^* spermatocytes (***Figure 7C***). The difference of expression between *Topbp1^+/+^* and *Topbp1^B5/B5^* was markedly higher in the SP2 to SP3 transition compared to the other transitions (***Figure 7D and E***) and was used to further split genes into three categories based on the severity of the silencing defect in *Topbp1^B5/B5^*: no defect (13 genes), mild (45 genes), or strong defect (170 genes) (***Figure 7F***). Notably, the severity of the silencing defect of a gene had some correlation with its RNA level in the pre-leptotene stage, with highly expressed genes having a higher tendency to have a more severe silencing defect (***Figure 7G***, ***Figure 7—figure supplement 2***). Taken together, these data characterize the specific silencing defect in *Topbp1^B5/B5^* spermatocytes and point to a specific role for TOPBP1 in ensuring proper silencing dynamics after an initial wave of MSCI, likely through later waves of silencing reinforcement. Our data is consistent with the notion that silencing of the X and Y chromosomes is a dynamic process that needs active and continuous engagement by the ATR-TOPBP1 signaling axis. Since the majority of the mouse models of male infertility accumulate pleiotropic defects, with disrupted MSCI and absence of sex body, the *Topbp1^B5/B5^* mouse reported here provides a unique model of DDR impairment in which MSCI can be uncoupled from sex body formation (***Figure 8***).

## Discussion

In male meiosis I, DDR factors such as ATR, TOPBP1, BRCA1, and the 9-1-1 complex play crucial roles in DNA repair, chromosome synapsis, recombination, sex body formation, and silencing (***Broering et al., 2014***; ***Ellnati et al., 2017***; ***Pacheco et al., 2018***; ***Pereira et al., 2022***; ***Royo et al., 2013***; ***Turner et al., 2004***; ***Widger et al., 2018***). Conditional depletion of these factors results in pleiotropic phenotypes from compound effects in multiple processes, with cells ultimately undergoing apoptosis-induced cell death during the pachytene checkpoint. Here, we report a mutant mouse model capable of deconvoluting TOPBP1's roles during meiosis I in males, separating its role in silencing from its roles in DNA repair, synapsis, and checkpoint signaling (***Figure 8***). While *Topbp1^B5/B5^* spermatocytes initiate XY silencing with similar dynamics as observed in *Topbp1^+/+^*, these cells fail to complete silencing at the final steps of MSCI. Of note, *Topbp1^B5/B5^* cells displayed slightly higher expression of X-linked genes than *Topbp1^+/+^* cells in the earlier spermatogenic stages (from spermatogonia 1 to spermatocyte 3). Furthermore, not all X-linked genes in *Topbp1^B5/B5^* spermatocytes were silenced; instead, some genes were only partially silenced while others exhibited increased expression after initial silencing.

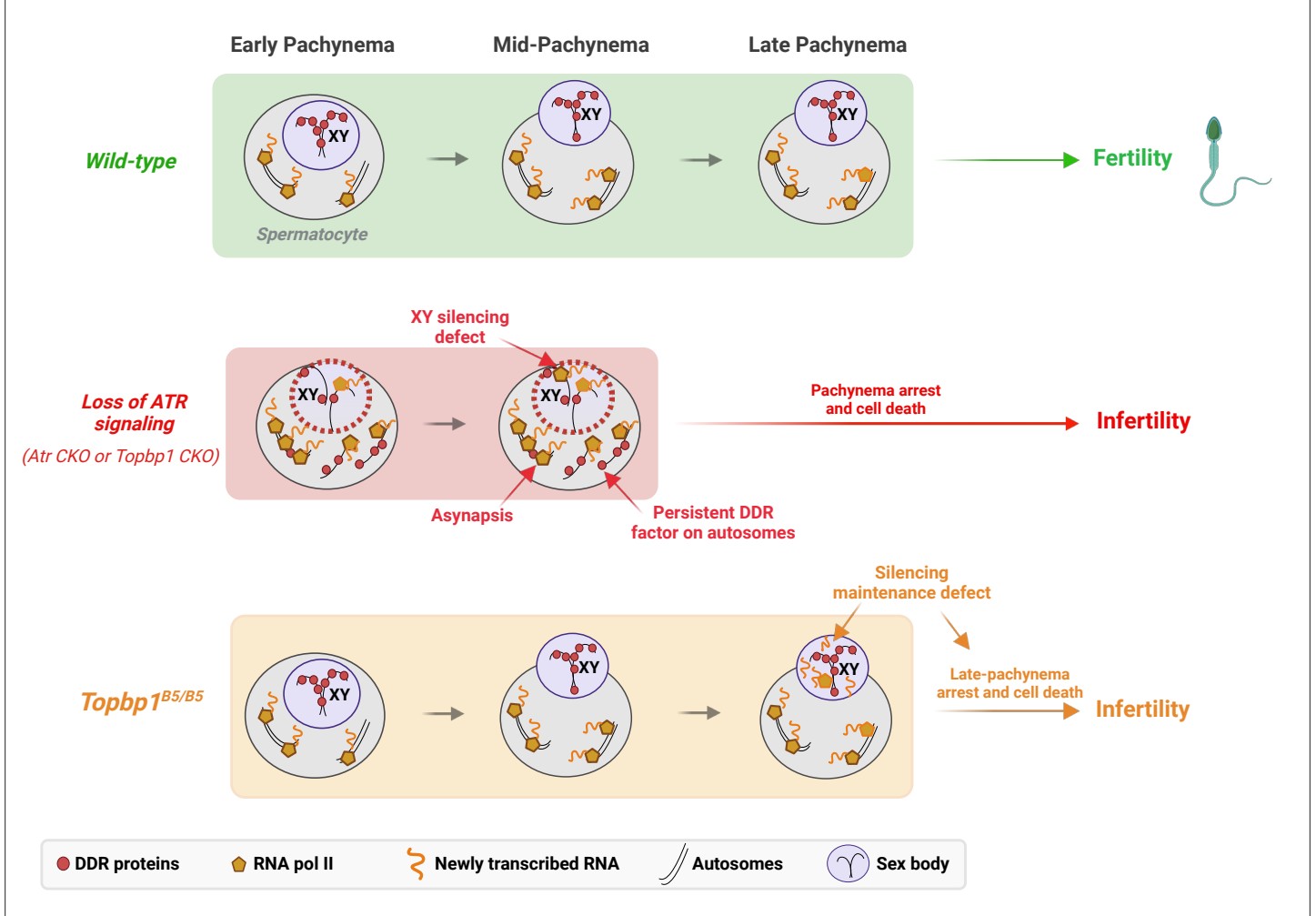

**Figure 8.** A new TOPBP1 mutant separates XY silencing from sex body formation. Schematic of sub-stages of meiotic prophase I. In wild-type mice, meiotic sex chromosome inactivation (MSCI) initiates following the accumulation of the DNA damage response (DDR) proteins at the XY chromosomes. During mid-pachytene, the XY body is fully formed, and transcription is restricted to the autosomes. In *Atr* or *Topbp1* CKOs, the sex body is not formed, and the DDR proteins are not sequestered to the XY. Asynapsis events and transcription of toxic genes at the sex chromosomes are observed, triggering mid-pachytene arrest. In *Topbp1^B5/B5^*, MSCI initiates, the sex body is normally formed with normal recruitment of DDR proteins to the X and Y chromosomes, yet cells fail in the reinforcement/maintenance of silencing. Cells progress through mid-pachytene but not into diplonema.

This is consistent with previous reports using mouse mutants with more severe MSCI defects, such as *Ago4^-/-^* and *Topbp1* CKO, in which not all X-linked genes exhibited altered levels of expression (*Ellnati et al., 2017*; *Modzelewski et al., 2012*). Interestingly, a sex body is formed that is morphologically indistinguishable from the sex body in wild-type animals. Several heterochromatin markers, as well as multiple canonical markers of sex body formation, localize properly in the sex body of *Topbp1^B5/B5^* mice. Overall, these findings suggest a non-canonical role for the ATR-TOPBP1 signaling axis in ensuring proper XY silencing dynamics during pachynema. This is the first DDR mutant that separates XY silencing from sex body formation, and that separates TOPBP1's role in spermatogenesis from its roles in organismal viability.

The *B5* allele reported here, which carries eight lysine to glutamic/aspartic acid substitutions in BRCT domain 5, is the first mutation shown to impair the meiotic silencing function of TOPBP1 in spermatocytes without severely disturbing TOPBP1's role in synapsis and sex body formation. Consistent with this being a separation of function mutant, *Topbp1^B5/B5^* males are viable and grossly normal, while completely sterile, whereas *Topbp1* null, or AAD mutated, mice are embryonic lethal (*Jeon et al., 2011*; *Zhou et al., 2013*). Moreover, depletion of TOPBP1 in mammalian cell lines triggers a robust G2/M arrest followed by cell senescence and loss of viability (*Jeon et al., 2011*) *Topbp1^B5/B5^*

MEFs do not display issues with cell proliferation or DDR defects. Based on these observations, and the finding of a silencing defect in *Topbp1^{B5/B5}* spermatocytes, it is likely that the role of TOPBP1 in silencing documented here could be specifically relevant in the context of male meiosis. It is important to note that other mutants previously reported to have XY silencing defects during meiotic prophase I, such as *Ago4* null and *Dicer* CKO, do not result in complete loss of sperm production, but a sub-fertility phenotype (*Greenlee et al., 2012*; *Modzelewski et al., 2012*). Therefore, we speculate that the specific type of silencing defect in *Topbp1^{B5/B5}* spermatocytes is particularly toxic, similar to other mutations in the DDR pathway that result in MSCI defects, which would explain the highly penetrant defect in sperm production.

The model that distinct TOPBP1 interactions mediate distinct ATR signaling pathways offers a potential explanation for why *Topbp1^{B5/B5}* have specific defects in silencing without noted effects in other key processes regulated by ATR, such as synapsis. In addition to binding and activating ATR through its AAD domain (*Mordes et al., 2008*; *Pereira et al., 2020*), TOPBP1 can bind to several proteins through its BRCT domains (*Yamane et al., 2002*) and act as a scaffolding protein to bring substrates in close proximity to ATR, thus facilitating the propagation of specific ATR signaling pathways. Experiments using ectopic expression of Flag-tagged TOPBP1 in HEK293T cells revealed that the set of mutations in *B5* disrupt the ability of TOPBP1 to interact with 53BP1 and BLM, as expected from previous reports (*Bigot et al., 2019*; *Blackford et al., 2015*; *Cescutti et al., 2010*; *Liu et al., 2017*; *Wang et al., 2013*). However, it is unclear whether the same interactions are also disrupted in spermatocytes or whether additional TOPBP1 interactions mediated by its BRCT5 specifically during meiotic prophase I are also disrupted. Apart from disrupting protein interactions, it is also possible that the observed changes in TOPBP1 protein stability in the *Topbp1 B5* mutant can contribute to impairing its roles in silencing. Such change in protein stability is consistent with a previous report showing that the TOPBP1-BLM interaction contributes to protein stabilization (*Balbo Pogliano et al., 2022*; *Wang et al., 2013*). Further work will be necessary to determine if the phenotypes observed in *Topbp1^{B5/B5}* spermatocytes are caused specifically by disruption of specific protein interactions or by a combination of disrupted interactions and reduced protein abundance. It is worth mentioning that the *Topbp1^{B5/B5}* phenotype is distinct from the *Topbp1* CKO despite the reduction in protein abundance.

Our finding that SETX localization to chromatin loops of the XY is impacted in *Topbp1^{B5/B5}* pachytene spermatocytes, together with our previous report that SETX undergoes ATR-dependent phospho-regulation in spermatocytes (*Sims et al., 2022*), suggest that an ATR-TOPBP1-SETX signaling axis is important for the silencing reinforcement in late MSCI. Genetic data support that impairment of this specific signaling axis would impact silencing without impacting synapsis. For example, *Topbp1* CKO, *Rad1* CKO, and *Atr* CKO spermatocytes display strong synapsis defect and defective entry in pachynema (*Ellnati et al., 2017*; *Widger et al., 2018*), whereas *Setx* null spermatocytes complete autosomal synapsis, while still displaying MSCI defects. On the other hand, is it likely that ATR signaling is controlling a specific aspect of SETX function since *Topbp1^{B5/B5}* spermatocytes do not share all defects observed in *Setx* null spermatocytes, as noted by the localization of γH2AX, sumoy-lation events, ubH2A and ATR at chromatin loops, which are defective in *Setx* null spermatocytes but normal in *Topbp1^{B5/B5}* spermatocytes. Moreover, *Topbp1^{B5/B5}* pachytene spermatocytes, but not *Setx* null spermatocytes, are able to reach the stage of crossover designation with MLH1 positive cells. Taken together, these observations suggest that *Topbp1^{B5/B5}* pachytene spermatocytes progress further in pachynema when compared to *Setx* null spermatocytes and are consistent with a model in which a ATR-TOPBP1 control only specific(s) mode of SETX regulation (*Pereira et al., 2022*; *Sims et al., 2022*).

The model involving SETX as a potential factor by which ATR controls silencing late in MSCI opens exciting directions to explore the interface of ATR-TOPBP1 with RNA processing. Given the established role for SETX in the resolution of R-loops (*Bennett and Spada, 2018*), it is tempting to speculate that silencing defects in *Topbp1^{B5/B5}* mutant may be related to aberrant accumulation of RNA-DNA hybrids that may affect removal of nascent mRNAs that is necessary for imposing silencing. This hypothesis assumes that the silencing of X and Y genes is a dynamic process involving ongoing mechanisms of exclusion of RNA polymerase II and nascent RNA, as proposed recently (*Abe et al., 2022*; *Sims et al., 2022*). Moreover, the model predicts that SETX function is specifically affected in the sex body, which is consistent with the observed defect in SETX localization. In support of this model, R-loops affect chromatin architecture at promoters and interfere with the recruitment of

transcription factors and chromatin remodelers, as observed in regions harboring CpG islands where R-loops prevent the action of DNA methyltransferases, thus preventing silencing (*Santos-Pereira and Aguilera, 2015*). Notably, highly transcribed genes, which tend to accumulate more R-loops (*Marnef and Legube, 2020*), displayed increased silencing defects in the *Topbp1^(B5/B5)* mutant.

While we have provided strong evidence to suggest a defect in later stages of MSCI as the cause of the cell death observed in *Topbp1^(B5/B5)* spermatocytes, we cannot exclude the potential contribution of other defects, beyond silencing, to the loss of diplotene cells. The increased number of MLH1 foci suggested an altered recombination pattern, possibly impairing the ratios of class I and class II cross-overs. The BRCT 5 domain of TOPBP1 interacts with the BLM helicase (*Balbo Pogliano et al., 2022*; *Blackford et al., 2015*), which has been found to play a role in meiotic recombination in yeast and mice (*Holloway et al., 2010*; *Rockmill et al., 2003*). *Blm* CKO mice display severe defects in prophase I progression in spermatocytes, including, incorrect pairing and synapsis of homologs, and defective processing of recombination intermediates, leading to increased chiasmata (*Holloway et al., 2010*). These observations raise the possibility that impaired meiotic progression and cell death in *Topbp1^(B5/B5)* spermatocytes is a combination of defects in MSCI and recombination. *Topbp1^(B5/B5)* spermatocytes do not progress beyond pachytene hence we were not able to visualize chiasmata and directly infer whether or not *Topbp1^(B5/B5)* is defective in crossing over. Of note, MLH1 and MLH3 are known to form a heterodimer in the context of meiotic recombination (*Lipkin et al., 2000*; *Svetlanov et al., 2008*). While our data show increased MLH1 foci counts in *Topbp1^(B5/B5)*, MLH3 foci counts were not different from *Topbp1^(+/+)*, thus, we cannot exclude the possibility that the imbalance between MLH1 and MLH3 might affect the loss of diplotene cells in *Topbp1^(B5/B5)* through processes not related to crossing over. Importantly, while MLH3 works exclusively as a heterodimer with MLH1, MLH1 can function in a heterodimeric complex with other MutL homologs (*Nakagawa et al., 1999*).

Our findings showing that TOPBP1 plays a specific role in silencing reinforcement after the first wave of MSCI are consistent with the recently proposed notion that the establishment and maintenance of MSCI is a dynamic process (*Abe et al., 2022*; *Sims et al., 2022*). Also consistent with this notion is our recent finding showing that ATR signaling is itself also highly dynamic and constantly being cycled (*Sims et al., 2022*). For example, using mice treated with the ATR inhibitor (ATRi) AZ20 for 4 hr, we found that such a short treatment is already sufficient to cause a complete loss of γH2AX, pMDC1, and SETX localization from the XY chromosomes (*Abe et al., 2022*; *Sims et al., 2022*). Furthermore, germ cells subjected to ATRi for 24 hr showed complete recovery of γH2AX only 3 hr after release from ATRi treatment (*Abe et al., 2022*). We propose that TOPBP1 acts on this phospho-cycle to ensure proper silencing reinforcement and maintenance, potentially by counteracting the engagement of anti-silencing factors that dynamically enter the sex body and need to be actively antagonized at the XY. Future work involving the characterization of possible unknown interactors of the BRCT domain 5 of TOPBP1, as well as functional dissection of ATR targets in MSCI, is essential to understand how TOPBP1 modulates the silencing machinery and shapes silencing dynamics. Interestingly, while we propose that the lack of silencing maintenance is the major defect causing the pachytene cell death in *Topbp1^(B5/B5)* spermatocytes, we cannot exclude the possibility that the expression of XY-linked genes could represent a regulated response to meiotic defects more than a mere consequence of a defective MSCI. If this latter hypothesis is true, the cell death caused by defects in the XY chromosomes would be independent of MSCI. Notably, this hypothesis could not have been conceptualized prior to this work given that the majority of prophase I mutants characterized to this date are unable to reach the stage where MSCI is properly established. Thus, the *Topbp1^(B5/B5)* is a unique model allowing future studies that may uncouple MSCI from XY-triggered cell death during pachynema.

In summary, our study presents a unique model for investigating the role of DDR factors in XY silencing. By allowing the uncoupling of MSCI progression from sex body formation, the *Topbp1^(B5/B5)* mutant enables the study of MSCI dynamics during key stages late in pachynema. Notably, the inability of *Topbp1^(B5/B5)* to sustain or reinforce silencing opens the possibility of uncovering new insights into the MSCI-dependent pachytene checkpoint.

# Materials and methods
## Materials availability
This study generated a unique antibody, RPA.

## Mice, genotyping, and treatment of mice with IR

All mice used in this work were handled following federal and institutional guidelines under a protocol approved by the Institutional Animal Care and Use Committee (IACUC protocol number 2011-0098) at Cornell University. CRISPR/Cas9 editing was used to engender the *Topbp1B5* allele and performed by the Cornell Mouse Modification core facility. To this end, the online tools CRISPR gold and Chop-Chop were used to generate high-quality (guide score >9) CRISPR guide RNAs targeting the intronic sequences neighboring the genomic sequence of *Mus musculus topbp1* exon 13. The CRISPR crRNAs (purchased from IDT) harboring the sequences cca<u>actcaggtcggccgctcttg</u> and cc<u>t</u>cgattagtcctcaagg cgag (PAM sites are underlined), both targeting the reverse strand, the repair template (below), and CAS9 RNA, were injected on embryos from super-ovulated, plugged C57BL/6J female mice crossed to C57BL/6J stud males. Two cell stage embryos were then implanted on pseudo pregnant females and pups were genotyped after 1 week old.

## Repair template

acagcagggcttctctgtgtaaccctctctccgtagaccagtttggccttgatcgaactcaggtcggccgctcttgcctctagagtcctggg attaaaggcgtgcactgccaccacccagagtatgtttctctgacattaaccatgctattattttttttaaaatgagctaattgtgtgttcatt tgctttatttccatgtaaaattttagTGTTCAAGAATTCTTTGTTGACGAAGCCAATGCAGAGGAAGGCATGCT CGCCAGCACACATCTTATAGTGAGGAACCGACTGGTTCCGAATACGAAGCTGCAGAGGAATGGA GTTTGCCGGCAGTTAACATTTCATGGCTCTTAGAAACTGCGGAAATCGGGGAGGAAGCAGATGA AAACCATTTTCTGGTTGACAACGCACCTAAACAAGgttagaagtccttgtttttttttttatgtattttacaacttgatgg tttctgaaataggatgttccagtacttgctttaaaacatttgtatgaccctaacctcagtcagtggtgcttacttcagaacccctgagtga aacacggaaagcagatcaatgaagaagcgcatcagggtcaacggtcgattagtcctcaaggcgagtgacgagaaggtgaccccccga atggctgttagaagcagttttttata (purchased from IDT as a G-block).

Intronic sequence is shown in lower case, exon 13 sequence is shown in capital letters, underline represents the mutated residues, blue represents the mutated PAM residues, purple shows the guide RNA sequences, and red shows the targeted amino acid sequences. Of note, although this repair template encodes for 11 amino acid changes, only 8 were successfully inserted into the mouse topbp1 exon 13 locus. For mice genotyping, the following primers were used: 5'-tgcatttccattaaccaacctc-3' and 5'- ggtagagttcaaatgtgtgtcatg-3' (also shown in Key Resources Table).

For irradiation, *Topbp1+/+* and *Topbp1B5/B5* mice were placed in a 137 cesium-sealed source irradiator (J.L. Shepherd and Associates) with a rotating turntable and irradiated with 7 Gy IR.

## MEFs and cell survival assays

MEFs were prepared from E13.5 mouse embryos as previously described (*Balmus et al., 2012*). Briefly, embryos were dissected and mechanically disrupted using pipette aspiration until homogeneous. Cells were allowed to settle, and the supernatant was transferred into Dulbecco's modified medium supplemented with 10% fetal bovine serum (FBS), 1% penicillin/streptomycin, and 1% nonessential amino acids. Following 4 days of growth, cells were then immortalized by transduction with a large-T antigen lentivirus. Subsequently, cells were selected with 10 µg/mL puromycin.

For colony survival assays, 500 cells were seeded per 10 cm dish, allowed to adhere for 24 hr, and treated with phleomycin or hydroxyurea for 24 hr (drug concentrations are displayed in *Figure 2A–D* and *Figure 2—figure supplement 1*). In the following day, cells were released and let to form colonies for 10–15 days. Cells were then washed once with PBS, fixed in 100% methanol for 1 hr, stained with 0.1% crystal violet (MP Biomedicals, 152511) solution overnight, and then washed with distilled water before imaging and counting.

For accessing DDR and checkpoint responses via western blot, $2 \times 10^6$ cells were seeded on a 60 cm dish, allowed to adhere for 24 hr, and treated with phleomycin or hydroxyurea for 3 hr (drug concentrations are displayed in *Figure 2*).

## Cell culture

HEK-293T cells were cultured in Dulbecco's modified medium supplemented with 10% fetal calf serum, 1% penicillin/streptomycin, and 1% nonessential amino acids. Immortalized MEFs were cultured in Dulbecco's modified medium supplemented with 10% fetal calf serum, 1% penicillin/streptomycin, 1% nonessential amino acids, and 1% glutamine supplementation. All cells were kept at 37°C and 5% $CO_2$. All the cell lines were regularly tested for mycoplasma contamination with the Universal

Mycoplasma Detection Kit (ATCC). HEK-293T cells were transfected using homemade polyethylenimine (Polysciences, Inc). Then, 36 hr after transfection, cells were treated with 1 mM HU (hydroxyurea) and then harvested for immunoprecipitation experiments.

## Plasmids

The full-length TOPBP1 CDS was cloned on a p3xflag vector (Milipore/Sigma E7658) using Gibson assembly (NEB) following the manufacturer's instructions. The p3xflag-TOPBP1 was used as a template to generate p3xflag-TOPBP-K155A, p3xflag-TOPBP-K250A, p3xflag-TOPBP-K704A, p3xflag-TOPBP-K1317A through site-directed mutagenesis using prime STAR master mix (Takara) and Gibson assembly to generate p3xflag-TOPBP-KE, using the following G-block (IDT) containing the eight charge-reversal point mutations at the BRCT 5 domain of *TOPBP1*:

AACGAATCCAATGCAGAAGAAGGCATGTTTGCCAGTACTCATCTTATACTGGAAGAACGTGGTG
GCTCTGAATATGAAGCTGCAAAGAAGTGGAATTTACCTGCCGTTACTATAGCTTGGCTGTTGGA
GACTGCTAGAACGGGAGAAGA.

All primers used for the cloning are shown in the Key Resources Table.

## Immunoblotting

Cells were harvested and lysed in modified RIPA buffer (50 mM Tris–HCl, pH 7.5, 150 mM NaCl, 1% tergitol, 0.25% sodium deoxycholate, 5 mM ethylenediaminetetraacetic acid [EDTA]) supplemented with complete EDTA-free protease inhibitor cocktail (Roche), 1 mM phenylmethylsulfonyl fluoride (PMSF), and 5 mM NaF. Whole-cell lysates, after sonication, were cleared by 15 min centrifugation at 17,000 × *g* at 4°C. Then, 20 µg of protein extract were mixed with 3× sodium dodecyl sulfate sample buffer and resolved by SDS-PAGE. Gels were transferred on polyvinylidene difluoride membranes and immunoblotted using standard procedures. Western blot signal was acquired with a Chemidoc Imaging System (Bio-Rad). Antibody information is provided in the Key Resources Table.

## Immunofluorescence

MEFs were grown on coverslips and then submitted to IR, 5 Gy and allowed to recover for 1.5 hr at 37°C and 5% $CO_2$. Cells were then fixed using 3.7% formaldehyde in phosphate-buffered saline (PBS) for 10 min at room temperature (RT). Fixed cells were then washed 3× with PBS, permeabilized for 5 min with 0.2% Triton X-100/PBS at RT and blocked in 10% bovine serum albumin/PBS for 20 min at RT. Coverslips were incubated first with primary antibodies for 2 hr at RT, followed by three washes with PBS, and then for 1 hr with relative secondary antibodies. After incubation with secondary antibodies, coverslips were washed three times with PBS and then mounted on glass microscope slides using DAPI–Vectashield mounting medium (Vector Laboratories). Slides were imaged on a Leica DMi8 Microscope with a Leica DFC9000 GTC camera using the LAS X (Leica Application Suite X) software with a ×100 objective. For micronuclei scoring, ~ 50 cells/replicate were counted. Two-tailed Student's *t*-test was used for statistical analysis. Antibody information is provided in the Key Resources Table.

## Meiotic spreads

Meiotic surface spreads were performed from 8- to 12-week-old mice as described by *Kolas et al., 2005*. Briefly, decapsulated testis from mice were incubated on ice in a hypotonic extraction buffer for 45 min. Tubules were then minced into single-cell suspension in 100 mM sucrose, and cells were spread on slides coated with 1% PFA with 0.15% TritionX- 100 and incubated in a humidifying chamber for 4 hr. For immunostaining, slides were blocked using 10% goat serum and 3% BSA, followed by incubation overnight with primary antibody (listed in the Key Resources Table) at 4°C in a humidifying chamber. Secondary antibodies were incubated at 37°C for 2 hr in the dark, and slides were then cover-slipped using antifade mounting medium (2.3% DABCO, 20 mM Tris pH 8.0, 8 µg DAPI in 90% glycerol). Slides were imaged on a Leica DMi8 Microscope with a Leica DFC9000 GTC camera using the LAS X software. For every condition, a minimum of 50 images from at least two independent mice were acquired. To quantify fluorescence intensity, the LAS X software quantification tool was used as previously described (*Sims et al., 2022*). Antibody information is provided in the Key Resources Table. p-Values were calculated in Prism–GraphPad using a linear mixed effect model (Nested *t*-test) that takes into account the variability in cells within each mouse when comparing mice between groups (*Topbp1*$^{+/+}$ vs *Topbp1*$^{B5/B5}$).

## 3D-structured illumination super-resolution microscope (3D-SIM)

Higher resolution images were acquired using an ELYRA 3D-structured illumination super-resolution microscopy (3D-SIM) from Carl Zeiss with ZEN Black software (Carl Zeiss AG, Oberkochen, Germany). Images are shown as maximum intensity projections of z-stack images. To reconstruct high-resolution images, raw images were computationally processed with ZEN Black. The brightness and contrast of images were adjusted using ImageJ (National Institutes of Health, USA). Antibody information is provided in the Key Resources Table.

## Fertility assays

For fertility testing, 8-week-old *Topbp1^B5/B5* females and C57BL/6 males or C57BL/6 females and *Topbp1^B5/B5* males were singly housed, where pregnancies were monitored for a period of 1 month. Viable pups were counted on the first day of life. For *Topbp1^B5/B5* males, breeding cages remained active for a period of 6 months at no time pregnant females nor birth of pups were detected. No noticeable defects were found on fertility of *Topbp1^B5/+*, males or females (data not shown).

## TUNEL

TUNEL assay was conducted using the Apoptag kit (EMD Millipore) following the manufacturer's instructions. The data were quantified in Image Scope by counting the number of positive cells per tubule for 100 tubules of each genotype, three mice each. Statistical differences between *Topbp1^+/+* and *Topbp1^B5/B5* were analyzed using Welch's unpaired $t$-test in GraphPad.

## Hematoxylin and eosin staining

Adult testes – from 12-week-old mice – were dissected and incubated in Bouin's fixative overnight, washed during 30 min each in 30%, then 50% and then 70% ethanol. The 70% ethanol wash was repeated three times more. Testes were then embedded in paraffin. 5 µm sections were mounted on slides. After rehydration in Safe Clear Xylene Substitute followed by decreasing amounts of ethanol, slides were stained with hematoxylin followed by eosin. The slides were then gradually dehydrated by incubation in increasing concentrations of ethanol before mounting using toluene mounting medium.

## Epididymal sperm counts

Caudal and epididymides from 8- to 12-week-old mice were minced with fine forceps at 37°C in a Petri dish containing Dulbecco's modified medium supplemented with 10% fetal calf serum, 1% penicillin/streptomycin, 4% BSA, and 1% nonessential amino acids. Samples were then incubated at 37°C for 30 min allowing sperm to swim out into the media, then fixed in 10% neutral-buffered formalin (1:25 dilution). Sperm cells were counted using a hemocytometer and analyzed statistically using two-tailed Student's $t$-test between *Topbp1^+/+* and *Topbp1^B5/B5*.

## Enrichment of testes phosphopeptides and TMT labeling

The enrichment of testes phosphopeptides and TMT labeling were done as described previously (*Sims et al., 2022*). Briefly, whole decapsulated testes were collected from 8- to 12-week-old mice after which tissue was subject to lysis, protein quantification, and normalization, denaturation, alkylation, precipitation, digestion, and solid-phase extraction (SPE) C18 cartridge clean up as described by *Sims et al., 2022*. Lyophilized tryptic peptides were then subject to phosphopeptide enrichment using a High-Select Fe-NTA Phosphopeptide Enrichment Kit according to the manufacturer's protocol (Cat# A32992, Thermo Scientific). Phosphopeptide samples were dried in silanized glass shell vials, resuspended in 50 mM HEPES, and labeled with 100 µg of TMT sixplex Isobaric Label Reagents (Thermo Scientific) using three TMT channels for each condition (*Topbp1^+/+* and *Topbp1^B5/B5*). The TMT-labeling reaction was done at RT for 1 hr and quenched with 5% hydroxylamine for 15 min. After quenching, TMT-labeled peptides from all six channels were pooled, acidified with 0.1% TFA, and desalted using a SPE 1cc C18 cartridge (Sep-Pak C18 cc vac cartridge, 50 mg Sorbent, WAT054955, Waters). Bound TMT-labeled phosphopeptides were eluted with 80% acetonitrile, 0.1% acetic acid in water before being dried via vacuum concentrator.

## Mass spectrometric analysis of TMT-labeled phosphopeptides

The dried TMT-labeled phosphopeptides were prefractionated using offline HILIC HPLC prior to being analyzed by mass-spectrometry as described by *Sims et al., 2022*. The LC-MS/MS was performed on

an UltiMate 3000 RSLC nano chromatographic system coupled to a Q-Exactive HF mass spectrometer (Thermo Fisher Scientific). The chromatographic separation was achieved via a 35-cm-long 100 μm inner diameter column packed in-house with 3 μm C18 reversed-phase resin (Reprosil Pur C18AQ 3 μm). The Q-Exactive HF was operated in data-dependent mode with survey scans acquired in the Orbitrap mass analyzer over the range of 380–1800 m/z with a mass resolution of 120,000. MS/MS spectra were performed after selecting the top 7 most abundant +2, +3, or +4 ions and a precursor isolation window of 0.7 m/z. Selected ions were fragmented by higher-energy collisional dissociation (HCD) with normalized collision energies of 28, with fragment mass spectra acquired in the Orbitrap mass analyzer with a monitored first mass of 100 m/z, mass resolution of 15,000, AGC target set to 1 × $10^5$, and maximum injection time set to 100 ms. A dynamic exclusion window of 30 s was specified.

## Phosphoproteomic data analysis

Trans Proteomic Pipeline (TPP) version 6.0.0 was used for phosphopeptide identification and quantification. MS data were converted to mzXML using msConvert as packaged with the TPP, after which spectral data files were searched using the Comet search engine (v2021 rev 1) (*Eng et al., 2013*). Peptide identifications were validated using PeptideProphet, phosphorylation site localization was performed using PTM Prophet, and TMT channel quantification was performed using Libra. Results from Libra were exported as tab-delimited files for further processing via R scripts as previously described (*Sims et al., 2022*). The mass spectrometry phosphoproteomics data have been deposited to the ProteomeXchange Consortium via the PRIDE (*Perez-Riverol et al., 2022*) a partner repository with the dataset identifier PXD042199.

## Immunoprecipitation

The immunoprecipitation (IP) experiments were performed as described by *Liu et al., 2017*. Briefly, cell pellets were lysed in 50 mM Tris–HCl, pH 7.5, 150 mM NaCl, 1% tergitol, 0.25% sodium deoxycholate, and 5 mM EDTA, supplemented with EDTA-free protease inhibitor cocktail, 5 mM sodium fluoride, 10 mM β-glycerolphosphate, 1 mM PMSF, and 0.4 mM sodium orthovanadate. The protein extracts were cleared by 10 min centrifugation and then incubated with anti-FLAG agarose beads (Sigma-Aldrich) for 16 hr at 4°C. The beads were then washed four times with the same buffer used for IP and then eluted using three resin volumes of the elution buffer (100 mM Tris–HCl, pH 8.0, and 1% SDS, and 1 mM DTT).

## Mass spectrometric analysis of immunoprecipitates

HEK-293T cells were grown in stable isotope labeling with amino acids in cell culture (SILAC) as previously described (*Liu et al., 2017*) and transfected as described above. Cells were treated with 1 mM HU for 16 hr before harvesting. Flag-TOPBP1 was immunoprecipitated using anti-FLAG agarose beads. Immunoprecipitates were then prepared for mass spectrometry analysis by reduction, alkylation, precipitation, and digestion by trypsin. The peptides were desalted, dried, and then fractionated by hydrophilic interaction chromatography as previously described (*Liu et al., 2017*). Fractions were dried and analyzed by liquid chromatography–tandem mass spectrometry using a mass spectrometer (Q-Exactive HF Orbitrap; Thermo Fisher Scientific). The capillary column was 35 cm long with a 100 μm inner diameter, packed in-house with 3 μm C18 reversed-phase resin (Reprosil Pur C18AQ 3 μm). Peptides were separated over an 70 min linear gradient of 6–40% acetonitrile in 0.1% formic acid at a flow rate of 300 nL/min as described previously (*Bastos de Oliveira et al., 2015*). Xcalibur 2.2 software (Thermo Fisher Scientific) was used for the data acquisition, and The Q-Exactive HF was operated in data-dependent mode with survey scans acquired in the Orbitrap mass analyzer over the range of 380–1800 m/z with a mass resolution of 120,000. The maximum ion injection time for the survey scan was 100 ms with a 3e6 automatic gain-control target ion. Tandem mass spectrometry spectra were performed by selecting up to the 20 most abundant ions with a charge state of 2, 3, or 4 and with an isolation window of 1.2 m/z. Selected ions were fragmented by higher energy collisional dissociation with a normalized collision energy of 28, and the tandem mass spectra were acquired in the Orbitrap mass analyzer with a mass resolution of 17,500 (at m/z 200). The TPP version 6.0.0 was used for peptide identification and SILAC quantification. MS data were converted to mzXML using msConvert as packaged with the TPP, after which spectral data files were searched using the Comet search engine (v2021 rev 1) (*Eng et al., 2013*). The following parameters were used in the

database search: semitryptic requirement, a mass accuracy of 15 ppm for the precursor ions, a differential modification of 8.0142 D for lysine and 10.00827 D for arginine, and a static mass modification of 57.021465 D for alkylated cysteine residues. Peptide identifications were validated using PeptideProphet and SILAC quantification was performed using XPRESS as described previously (*Bastos de Oliveira et al., 2015*; *Sims et al., 2022*). The mass spectrometry data have been deposited to the ProteomeXchange Consortium via the PRIDE (*Perez-Riverol et al., 2022*) partner repository with the dataset identifier PXD042199.

## Total germ cells preparation

Mice testis were collected from 8- to 12-week-old mice (n = 5 mice, 10 testis for *Topbp1*$^{+/+}$, and n = 20–30 mice, 40–60 testis for *Topbp1*$^{B5/B5}$), and dissociated using standard protocols for germ cell extraction (*Grive et al., 2019*). Briefly, decapsulated testes were held in 10 mL of preheated (35°C) DMEM-F12 buffer containing 2 mg of Collagenase 1A and DNAse 7 mg/mL, on a 50 mL conical tube. The collagenase digestion was performed at a 35°C shaker water bath, 150 rpm during 5 min. The collagenase digestion was stopped by the addition of 40 mL of DMEM-F12 and the tubules were let to decantate for 1 min. The supernatant was removed and added another 40 mL of DMEM-F12 to further wash the residual collagenase and remove somatic and excessive sperm cells. Next, the tubules were digested using 10 mL of DMEM-F12 buffer containing 5 mg of trypsin (Thermo Fisher 27250018) on a 50 mL conical tube and the reaction was carried out on a 35°C shaker water-bath, 150 rpm during 5 min. Digested tubules were strained on a 100 µm strainer containing 3 mL of FBS (100% FBS, heat-inactivated, Sigma F4135-500mL). Total germ cells were centrifuged at 300 × *g* for 5 min, 4°C and checked for single cells and viability.

## Flow cytometry analysis

Total germ cell extracts were stained with Vybrant dye cycle (VDG) (Invitrogen) 100 µM for 30 min at RT, kept on dark, and rocking. After staining, cells were sorted as previously described (*Rodríguez-Casuriaga et al., 2014*) aiming to enrich for spermatocytes and using the Sony MA900 fluorescent-activated cell sorter (FACS), tuned to emit at 488 nm and with a 100 µm nozzle. Laser power was set to collect VDG-emitted fluorescence in FL1. Sorted cells were collected on 1.5 mL tubes containing 0.5 mL of DMEM-F12 buffer + 10% FBS. The FACS was done at the Flow Cytometry Facility, Cornell University.

## Single-cell RNA sequencing

Flow-sorted cells were submitted to the Cornell DNA Sequencing Core Facility and processed on the 10X Genomics Chromium System targeting 5000–7000 cells per sample as described previously (*Grive et al., 2019*) using the 10X Genomics Chromium Single Cell 3′ RNA-seq v2 kit to generate the sequencing libraries, which were then tested for quality control on an ABI DNA Fragment Analyzer and ran on a NextSeq platform with 150 base-pair reads. The sequencing was carried out to an average depth of 98M reads (range 77–124M); on average, 91% of reads (range 89%–92%) and then mapped to the reference genome.

## Single-cell RNA sequencing data analysis

Count matrices were generated for each sample using the Cell Ranger counts function and then imported into Seurat and integrated. Cells were filtered based on gene number, UMI, counts, and mitochondrial percentage. Cells with less than 500 genes, less than 1000 UMIs, or more than 5% of mitochondrial reads were excluded from further analysis (*Figure 6—figure supplement 16*). Raw counts were normalized using Seurat NormalizeData using default parameters and the top 4000 variable features were identified using the FindVariableFeatures function using ExpMean for the mean function and LogVMR for the dispersion function. Principal components were calculated from variable genes and used with Harmony to correct for batch effect. Harmony embeddings dimensions 1–20 were used for a Shared Nearest Neighbor graph with k = 30, unsupervised clustering with a resolution of 4, and Uniform Manifold Approximation and Projection (UMAP) analysis. Cell types were manually identified by use of marker genes, and the SingleR package was used to confirm cell identity. Chromosome X/autosome 9 ratios were calculated by taking the mean gene expression of all genes on chromosome 9 or X for a cell and dividing it by the mean gene expression for all autosomal genes in

a cell. Ratios were visualized using ggplot and the introdataviz package geom_split_violin function. The single-cell heatmap was generated using the DoHeatmap function. The clustered analysis of the X-linked genes shown in *Figure 7A* was done by splitting the detected 233 X-linked genes into three distinct categories based on the difference of RNA level between the spermatocyte stages and normalized by pre-leptotene: reduced silencing (>1), unaltered silencing ($\geq$ –1 or $\leq$1), or increased silencing (< –1). To further split the X-linked genes into three categories based on the severity of the silencing defect (detected at spermatocyte 3 stage) in *Topbp1*$^{B5/B5}$ – no defect (13 genes), mild (45 genes), or strong defect (170 genes) – the difference in expression between SP3 genes from *Topbp1*$^{+/+}$ or from *Topbp1*$^{B5/B5}$ was calculated. The categories were defined as strong > 2.5, mild $\geq$ 1 or $\leq$ 2.5, and no defect < 1. The single-cell RNAseq data generated in this study have been deposited in the NCBI GEO database under accession number GSE232588.

## Acknowledgements

We thank the members of Cohen lab, Weiss lab, Grimson lab, and Smolka lab for comments and suggestions. We thank Dr. Fenghua Hu and Dr. Tony Bretscher for the use of the microscopes and Beatriz Almeida for technical assistance. We thank Dr. Lydia Tesfa and the BRC Flow Cytometry Facility (RRID:SCR_021740) for training and assistance with FACS at the Cornell Institute of Biotechnology, Peter Schweitzer and the BRC Genomics Facility (RRID:SCR_021727) at the Cornell Institute of Biotechnology for sequencing experiments, Johana M de la Cruz and the BRC Imaging Facility (RRID:SCR_021741) at the Cornell Institute of Biotechnology for training and assistance with the Elyra Super Resolution Microscope and the 3D-SIM imaging. We thank Marco Hiler and the BMS and Stem Cell Pathology Unit at the Cornell Veterinary Research Tower for training and assistance with imaging histological sections using ScanScope. We thank Dr. John Schimenti, Dr. Robert Munroe, Mr. Christian Abratte, and Cornell's Stem Cell and Transgenic Core Facility for generating the *Topbp1*$^{B5/B5}$ mouse. We thank Dr. Mary Ann Handel for the H1T antibody. This work was supported by the Eunice Kennedy Shriver National Institute of Child Health and Human Development (R01-HD095296) to MBS and RW, National Institute of Health – General Medicine (R35-GM141159) to MBS, National Institute of Health (R01HD041012) to PEC, the Eunice Kennedy Shriver National Institute of Child Health and Human Development (P50HD104454) to AG, the Spanish Agencia Estatal de Investigación (PID2019-109222RB-I00/AEI/10.13039/501100011033); European Union Regional Funds (FEDER) to RF, and 2022 CVG Scholar Award from the Center of Vertebrates Genomics and the 2022 ASCB International training Scholarship to CFRA.

## Additional information

### Funding

| Funder | Grant reference number | Author |
| --- | --- | --- |
| Eunice Kennedy Shriver National Institute of Child Health and Human Development | R01-HD095296 | Robert S Weiss<br>Marcus B Smolka |
| National Institute of General Medical Sciences | R35-GM141159 | Marcus B Smolka |
| Eunice Kennedy Shriver National Institute of Child Health and Human Development | R01HD041012 | Paula E Cohen |
| Eunice Kennedy Shriver National Institute of Child Health and Human Development | P50HD104454 | Andrew Grimson |
| Spanish Agencia Estatal de Investigación | PID2019-109222RB-877 I00/AEI/10.13039/501100011033 | Raimundo Freire |

| Funder | Grant reference number | Author |
| --- | --- | --- |
| European Union Regional Funds | | Raimundo Freire |
| 2022 CVG Scholar Award from the Center of Vertebrates Genomics | | Carolline Ascenção |
| 2022 ASCB International training Scholarship | | Carolline Ascenção |

The funders had no role in study design, data collection and interpretation, or the decision to submit the work for publication.

## Author contributions

Carolline Ascenção, Conceptualization, Resources, Data curation, Formal analysis, Funding acquisition, Validation, Investigation, Visualization, Methodology, Writing – original draft, Writing – review and editing; Jennie R Sims, Conceptualization, Data curation, Formal analysis, Validation, Investigation, Visualization, Methodology; Alexis Dziubek, Data curation, Software, Formal analysis, Methodology; William Comstock, Jumana Badar, Data curation; Elizabeth A Fogarty, Data curation, Methodology; Raimundo Freire, Resources, Funding acquisition; Andrew Grimson, Data curation, Formal analysis, Funding acquisition, Methodology; Robert S Weiss, Conceptualization, Resources, Formal analysis, Supervision, Funding acquisition, Project administration, Writing – review and editing; Paula E Cohen, Conceptualization, Resources, Data curation, Formal analysis, Supervision, Funding acquisition, Validation, Investigation, Methodology, Writing – original draft, Project administration, Writing – review and editing; Marcus B Smolka, Conceptualization, Resources, Data curation, Formal analysis, Supervision, Funding acquisition, Validation, Investigation, Visualization, Methodology, Writing – original draft, Project administration, Writing – review and editing

## Author ORCIDs

Carolline Ascenção http://orcid.org/0000-0002-0882-7390
Raimundo Freire http://orcid.org/0000-0003-4473-8894
Robert S Weiss http://orcid.org/0000-0003-3327-1379
Paula E Cohen https://orcid.org/0000-0002-2050-6979
Marcus B Smolka http://orcid.org/0000-0001-9952-2885

## Ethics

This study was performed in strict accordance with the recommendations in the Guide for Care and Use of Laboratory Animals of the National Institute of Health. All of the animals were handled according to approved institutional animal care and use committee (IACUC) protocol 2011-0098 of Cornell University.

Reviewer #1 (Public Review): https://doi.org/10.7554/eLife.90887.3.sa1
Reviewer #2 (Public Review): https://doi.org/10.7554/eLife.90887.3.sa2
Reviewer #3 (Public Review): https://doi.org/10.7554/eLife.90887.3.sa3
Author Response https://doi.org/10.7554/eLife.90887.3.sa4

# Additional files

## Supplementary files

• MDAR checklist

## Data availability

The mass spectrometry data generated in this study have been deposited to the ProteomeXchange Consortium via the PRIDE (*Perez-Riverol et al., 2022*) partner repository with the dataset identifier PXD042199.The Single-cell RNAseq data generated in this study have been deposited in the NCBI GEO database under accession number GSE232588.

The following datasets were generated:

| Author(s) | Year | Dataset title | Dataset URL | Database and Identifier |
|---|---|---|---|---|
| Ascencao CFR, Sims JR, Dziubek A, Comstock W, Fogarty EA, Badar J, Freire R, Grimson A, Weiss RS, Cohen PE, Smolka MB | 2024 | TOPBP1 Shapes XY Silencing Dynamics Independently of Sex Body Formation to Ensure Male Fertility | https://proteomecentral.proteomexchange.org/cgi/GetDataset?ID=PXD042199 | ProteomeXchange, PXD042199 |
| Ascencao CFR, Sims JR, Dziubek A, Comstock W, Fogarty EA, Badar J, Freire R, Grimson A, Weiss RS, Cohen PE, Smolka MB | 2024 | TOPBP1 Shapes XY Silencing Dynamics during Meiosis to Ensure Male Fertility | https://www.ncbi.nlm.nih.gov/geo/query/acc.cgi?acc=GSE232588 | NCBI Gene Expression Omnibus, GSE232588 |

The following previously published dataset was used:

| Author(s) | Year | Dataset title | Dataset URL | Database and Identifier |
|---|---|---|---|---|
| Sims JR, Faca VM, Pereira C, Ascencao C, Comstock W, Badar J, Arroyo-Martinez GA, Freire R, Cohen PE, Weiss RS, Smolka MB | 2021 | Phosphoproteomics of ATR Signaling in Prophase I of Mouse Meiosis | https://proteomecentral.proteomexchange.org/cgi/GetDataset?ID=PXD023803 | ProteomeXchange, PXD023803 |

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

## Appendix 1

**Appendix 1—key resources table**

| Reagent type (species) or resource | Designation | Source or reference | Identifiers | Additional information |
|---|---|---|---|---|
| Sequence-based reagent | p3xflag-TOPBP1-Flag-N-terminal-F | IDT | PCR primer | attcatcgatagatctgataAT GTCCAGAAATGACAAAGA |
| Sequence-based reagent | p3xflag-TOPBP1-Flag-N-terminal-R | IDT | PCR primer | tagagtcgactggtaccgatttag TGTACTCTAGGTCGTT |
| Sequence-based reagent | *TOPBP1*_K1317A_R | IDT | Site-directed mutagenesis | tgccactgaggctaaatacg cctcgtttcgaagtggatgt |
| Sequence-based reagent | *TOPBP1*_K1317A_F | IDT | Site-directed mutagenesis | acatccacttcgaaacgagg cgtatttagcctcagtggca |
| Sequence-based reagent | *TOPBP1*_K155A_F | IDT | Site-directed mutagenesis | caggtttgcagcaactaaat atgctttgctaccaacttctcctgca |
| Sequence-based reagent | *TOPBP1*_K155A_R | IDT | Site-directed mutagenesis | tgcaggagaagttggtagcaaa gcatatttagttgctgcaaacctg |
| Sequence-based reagent | *TOPBP1*_K250A_R | IDT | Site-directed mutagenesis | tctcttggcacactcatacg cctgaccttttggttcttgc |
| Sequence-based reagent | *TOPBP1*_K250A_F | IDT | Site-directed mutagenesis | gcaagaaccaaaaggtcagg cgtatgagtgtgccaagaga |
| Sequence-based reagent | *Topbp1*_BRCT5-Fwd genotyping | IDT | PCR primer | tgcatttccattaaccaacctc |
| Sequence-based reagent | *Topbp1*_BRCT5 Rev genotyping | IDT | PCR primer | ggtagagttcaaatgtgtgtcatg |
| Antibody | Anti-phospho H2A.X (Ser139) (mouse monoclonal) | Millipore | Cat# 05-636; RRID:AB_309864 | IF (meiotic spreads) 1:50,000 IF 1:2000 |
| Antibody | Anti-SCP3 antibody (mouse monoclonal) | Abcam | Cat# ab97672; RRID:AB_10678841 | IF (meiotic spreads) 1:1000 |
| Antibody | Anti-SYCP3 (rabbit polyclonal) | *Lenzi et al., 2005* | Custom | IF (meiotic spreads) 1:10,000 |
| Antibody | Anti-SCP1 (rabbit polyclonal) | Abcam | ab15090 | IF (meiotic spreads) 1:1000 |
| Antibody | Anti-Rad51 (rabbit polyclonal) | Millipore | PC130 | IF (meiotic spreads) 1:1000 |
| Antibody | Anti-ATR (rabbit polyclonal) | Cell signaling | 2790 | IF (meiotic spreads) 1:1000 |
| Antibody | Anti-TOPBP1 (rabbit polyclonal) | *Danielsen et al., 2009* | Custom | IF (meiotic spreads) 1:500 Western blot 1:1000 |
| Antibody | Anti-phospho-Chk1 (ser317) (rabbit monoclonal) | Cell Signaling | 12302 | IF (meiotic spreads) 1:100 Western blot 1:1000 |
| Antibody | Anti-H1T (guinea pig polyclonal) | A gift from Dr. Mary Ann Handel; *Inselman et al., 2003* | Custom | IF (meiotic spreads) 1:500 |
| Antibody | Anti-HORMAD2 (rabbit polyclonal) | *Wojtasz et al., 2012* | Custom | IF (meiotic spreads) 1:500 |
| Antibody | Anti-HORMAD1 (rabbit polyclonal) | *Wojtasz et al., 2012* | Custom | IF (meiotic spreads) 1:500 |

*Appendix 1 Continued on next page*

*Appendix 1 Continued*

| Reagent type (species) or resource | Designation | Source or reference | Identifiers | Additional information |
|---|---|---|---|---|
| Antibody | Phospho HORMAD2 (S271) (rabbit polyclonal) | *Wojtasz et al., 2009* | Custom | IF (meiotic spreads) 1:500 |
| Antibody | Anti-GAPDH (mouse monoclonal) | Thermo Fisher Scientific | AM4300 | Western blot 1:5000 |
| Antibody | Anti-β-actin (rabbit polyclonal) | Cell Signaling | 4967 | Western blot 1:5000 |
| Antibody | Anti-phospho MDC1 (T4) (rabbit polyclonal) | Abcam | Ab35967 | IF (meiotic spreads) 1:500 |
| Antibody | Anti-SETX (rabbit polyclonal) | Abcam | Ab220827 | IF (meiotic spreads) 1:100 |
| Antibody | Anti-MLH1 (mouse monoclonal) | BD Biosciences | 550838 | IF (meiotic spreads) 1:200 |
| Antibody | Anti-MLH3 (guinea pig polyclonal) | *Holloway et al., 2014* | Custom | IF (meiotic spreads) 1:200 |
| Antibody | Anti-CHK1 (mouse monoclonal) | Santa Cruz | sc-8408 | Western blot 1:1000 |
| Antibody | Anti-CHK2 (mouse monoclonal) | Millipore | 05-649 (clone7) | Western blot 1:1000 |
| Antibody | Anti-phospho CHK1 (S345) (rabbit polyclonal) | Cell Signaling | 2341 | IF (meiotic spreads) 1:200 Western blot 1:1000 |
| Antibody | Anti-phospho RPA (S4/S8) (rabbit polyclonal) | Bethyl | A300-245A | Western blot 1:1000 |
| Antibody | RPA (made against full length RPA2 expressed and purified in *E. coli,* and injected to rabbit) (rabbit polyclonal) | N/A | Custom | Western blot 1:1000 |
| Antibody | Anti-KAP1 (rabbit monoclonal) | Bethyl | a700-014-T | Western blot 1:1000 |
| Antibody | Anti-phospho KAP1 (S824) (rabbit polyclonal) | Bethyl | A300-767A-T | Western blot 1:1000 |
| Antibody | Anti-53BP1 (rabbit polyclonal) | Cell Signaling | 4937 | Western blot 1:1000 |
| Antibody | Anti-53BP1 (rabbit polyclonal) | Novus Biologicals | NB100-304 | IF (meiotic spreads) 1:200 |
| Antibody | Anti-BLM (rabbit polyclonal) | Abcam | ab2179 | Western blot 1:500 |
| Antibody | Anti-Rad9 (rabbit polyclonal) | Bethyl | A300-890A-T | Western blot 1:1000 |
| Antibody | Anti-BRCA1 (rabbit polyclonal) | *Kakarougkas et al., 2013* | Custom | IF (meiotic spreads) 1:200 |
| Antibody | Anti-BRCA1 (rabbit polyclonal) | Proteintech | 22362-1-AP | Western blot 1:1000 |
| Antibody | Anti-MDC1 (rabbit polyclonal) | *Modzelewski et al., 2015* | Custom | IF (meiotic spreads) 1:200 IF 1:200 |
| Antibody | Anti-RNA pol 2 (mouse monoclonal) | Millipore | 05-623 | IF (meiotic spreads) 1:2000 |

*Appendix 1 Continued on next page*

*Appendix 1 Continued*

| Reagent type (species) or resource | Designation | Source or reference | Identifiers | Additional information |
|---|---|---|---|---|
| Antibody | Anti-phospho RNA pol 2 (Ser2) (rat monoclonal) | Millipore | 04-1571 | IF (meiotic spreads) 1:400 |
| Antibody | Anti-phospho RNA pol 2 (Ser5) (rat monoclonal) | Millipore | 04-1572 | IF (meiotic spreads) 1:400 |
| Antibody | Anti-H3K9ac (rabbit polyclonal) | Abclonal | A7255 | IF (meiotic spreads) 1:200 |
| Antibody | Anti-H3K9me3 (rabbit polyclonal) | Active Motif | 39766 | IF (meiotic spreads) 1:200 |
| Antibody | Anti-H3K4me1 (rabbit polyclonal) | Abclonal | A2355 | IF (meiotic spreads) 1:200 |
| Antibody | Anti-CHD4 (rabbit polyclonal) | Abclonal | A10557 | IF (meiotic spreads) 1:200 |
| Antibody | Anti-Sumo_2/3 (rabbit polyclonal) | Proteintech | 11251-1-AP | IF (meiotic spreads) 1:200 |
| Antibody | Anti-USP7 (mouse monoclonal) | Proteintech | 66514-1-Ig | IF (meiotic spreads) 1:200 |
| Antibody | Anti-H3K27ac (rabbit polyclonal) | Active Motif | 39134 | IF (meiotic spreads) 1:200 |
| Antibody | Anti-H3K36ac (rabbit polyclonal) | Active Motif | 61102 | IF (meiotic spreads) 1:200 |
| Antibody | Anti-H3K4me3 (rabbit polyclonal) | Active Motif | 39160 | IF (meiotic spreads) 1:200 |
| Antibody | Anti-H4K16ac (rabbit polyclonal) | Abclonal | A5280 | IF (meiotic spreads) 1:200 |
| Antibody | Anti-H4c (rabbit polyclonal) | Millipore | 06-598 | IF (meiotic spreads) 1:200 |
| Antibody | Anti-H2AK199ub (rabbit monoclonal) | Abcam | Ab193203 | IF (meiotic spreads) 1:200 |
| Antibody | Goat anti-rabbit IgG (H+L) highly cross-adsorbed secondary antibody, Alexa Fluor 488 (goat polyclonal) | Thermo Fisher Scientific | A-11034 | IF (meiotic spreads) 1:1000 |
| Antibody | Goat anti-mouse IgG (H+L) antibody, Alexa Fluor 488 conjugated (goat polyclonal) | Thermo Fisher Scientific | A-11017 | IF (meiotic spreads) 1:1000 |
| Antibody | Goat anti-rabbit IgG (H+L) antibody, Alexa Fluor 594 conjugated (goat polyclonal) | Thermo Fisher Scientific | A-11012 | IF (meiotic spreads) 1:1000 |
| Antibody | Goat anti-mouse IgG (H+L) highly cross-adsorbed secondary antibody, Alexa Fluor Plus 594 (goat polyclonal) | Thermo Fisher Scientific | A32742 | IF (meiotic spreads) 1:1000 |
| Antibody | Goat anti-guinea pig IgG (H+L) highly cross-adsorbed secondary antibody, Alexa Fluor 647 (goat polyclonal) | Thermo Fisher Scientific | A-21450 | IF (meiotic spreads) 1:1000 |
| Commercial assay or kit | ApopTag Plus Peroxidase In Situ Apoptosis Kit | Millipore | S7101 | |

*Appendix 1 Continued on next page*

*Appendix 1 Continued*

| Reagent type (species) or resource | Designation | Source or reference | Identifiers | Additional information |
|---|---|---|---|---|
| Software, algorithm | GraphPad Prism 9 | GraphPad Prism | RRID:SCR_002798 | |
| Software, algorithm | Cellranger v7.0.0 | 10X Genomics | https://support.10xgenomics.com/single-cell-gene-expression/software/downloads/3.1 | |
| Software, algorithm | R v4.2.1 | R Foundation | RRID:SCR_001905 | |
| Software, algorithm | Seurat v4.3.0 | *Hao et al., 2021* | RRID:SCR_007322 | |
| Software, algorithm | sva v3.44.0 | *Leek et al., 2023* | RRID:SCR_012836 | |
| Software, algorithm | SingleR v1.10.0 | *Aran et al., 2019* | RRID:SCR_023120 | |
| Software, algorithm | Reshape2 v1.4.4 | *Wickham, 2007* | RRID:SCR_022679 | |
| Software, algorithm | biomaRt v2.52.0 | *Durinck et al., 2009* | RRID:SCR_019214 | |
| Software, algorithm | ggpubr v0.5.0 | *Kassambara, 2019* | RRID:SCR_021139 | |
| Software, algorithm | Tidyverse v1.3.2 | *Wickham et al., 2019* | RRID:SCR_019186 | |
| Software, algorithm | ggnewscale v0.4.8 | Campitelli | https://eliocamp.github.io/ggnewscale/index.html | |
| Software, algorithm | Harmony v0.1.1 | *Korsunsky et al., 2019* | RRID:SCR_022206 | |
| Software, algorithm | Org.Mm.eg.db v3.15.0 | *Carlson et al., 2019* | RRID:SCR_023488 | |
| Software, algorithm | Corrplot v0.92 | *Wei and Simko, 2017* | RRID:SCR_023081 | |
| Software, algorithm | SeuratWrappers v0.3.1 | Satija Lab | RRID:SCR_022555 | |
| Software, algorithm | gghalves v0.1.4 | *Tiedemann, 2022* | https://github.com/erocoar/gghalves | |

