## [Editor Report · eLife assessment]

This **important** study reports a new mutant mouse line with compromised function of a DNA damage response protein. The evidence supporting the conclusion that the mutants display defective maintenance of meiotic sex chromosome inactivation is **solid**. This work is of interest to biomedical researchers working on meiosis and meiotic sex chromosome inactivation.

---

## [Referee Report · Reviewer #1 (Public Review)]

This is a very well-written and performed study describing a TOPBP1 separation of function mutation, resulting in defective MSCI maintenance but normal sex body formation. The phenotype differs from that of a previous TOPBP1 null allele, in which both MSCI and sex body formation were defective. Additional defects in CHK phosphorylation and SETX localization are also described.

Strengths:

The study is very rigorous, with a remarkably large number spectrum of techniques deployed to support the conclusions

Weaknesses

The study claims that MSCI is initiated but not maintained in the mutant. I think alternative hypotheses are possible.

---

## [Referee Report · Reviewer #2 (Public Review)]

Summary:

This paper described the role of BRCT repeat 5 in TOPBP1, a DNA damage response protein, in the maintenance of meiotic sex chromosome inactivation (MSCI). By analyzing a Topbp1 mutant mouse with amino acid substitutions in BRCT repeat 5, the authors found reduced phosphorylation of a DNA/RNA helicase, Sentaxin, and decreased localization of the protein to the X-Y sex body in pachynema. Moreover, the authors also found decreased repression of several genes on the sex chromosomes in the male mice.

Strengths:

The works including phospho-proteomics and single-cell RNA sequencing with lots of data have been done with great care and most of the results are convincing.

Weaknesses:

No weakness.

---

## [Referee Report · Reviewer #3 (Public Review)]

The work presented by Ascencao and coworkers aims to deepen into the process of sex chromosome inactivation during meiosis (MSCI) as a critical factor in the regulation of meiosis progression in male mammals. For this purpose, they have generated a transgenic mouse model in which a specific domain of TOPBP1 protein has been mutated, hampering the binding of a number of protein partners and interfering with the regulatory cascade initiated by ATR. Through the use of immunolocalization of an impressive number of markers of MSCI, phosphoproteomics and single cell RNA sequencing (scRNAseq), the authors are able to show that despite a proper morphological formation of the sex body and the incorporation of most canonical MSCI makers, sex chromosome-liked genes are reactivated at some point during pachytene and this triggers meiosis progression breakdown, likely due to a defective phosphorylation of the helicase SETX.

The manuscript presents a clear advance in the understanding of MSCI and meiosis progression with two main strengths. First, the generation of a mouse model with a very uncommon phenotype. Second, the use of a vast methodological approach. The results are well presented and illustrated. Nevertheless, the discussion could be still a bit tuned by the inclusion of some ideas, and perhaps speculations, that have not been considered.

---

## [Author Response]

The following is the authors’ response to the original reviews.

**Public Reviews:F**

**Reviewer #1 (Public Review):**
Summary:This is a very well written and performed study describing a TOPBP1 separation of function mutation, resulting in defective MSCI maintenance but normal sex body formation. The phenotype differs from that of a previous TOPBP1 null allele, in which both MSCI and sex body formation were defective. Additional defects in CHK phosphorylation and SETX localization are also described.Strengths:The study is very rigorous, with a remarkably large number of MSCI marks assayed, phosphoproteomics (leading to the interesting SETX discovery) and 10X RNAseq, allowing the MSCI phenotype to be further deconvolved. The approaches in most cases are robust.Weaknesses:There aren't many; please find list below:1. The authors are committed to the idea that maintenance of MSCI is the major defect here. However, based on the data, an alternative would be that some cells achieve sex body formation and MSCI normally, while others do not. It would only take a small percentage of cells exhibiting MSCI failure to kill all the cells in the same germinal epithelium, so this could still explain the complete pachytene block. This isn't a major point...this phenotype is clearly different to the TOPBP1 KO, but a broader discussion of possibilities in the discussion would help. I raise this in the context of both the cytology and 10X analysis:(a) The assessment that sex body formation is normal is based on cytology in Figure 3-figure supplement 3 and Figure 3-figure supplement 4, but a more rigorous approach would be to assess condensation of the XY pair in stage-matched spread cells (maybe they have that data already) by measuring distances between the X and Y centromere, or looking at stage IV of the seminiferous cycle, where all cells should have oval sex bodies but sex body mutants have persistent elongated XY pairs (see work of Namekawa and Turner). The authors do actually mention that gH2AX spreading is defective in many cells....and if this is true, condensation to form a sex body would almost certainly not have taken place in those cells.

We appreciate the reviewer’s comment and have performed the experiment suggested, counting the number of elongated sex bodies in all sex body-positive cells in seminiferous tubules stained with γH2AX and DAPI (as done by Turner in Hirota et al., 2018). The experiment did not show significant differences between *Topbp1^+/+^* and *Topbp1^B5/B5^* as shown in Author response image 1.

**Author response image 1. sa4fig1:** *Topbp1^B5/B5^* displays normal condensation of the XY-pair. (A) Immunostaining of XY condensation in *Topbp1^+/+^* and *Topbp1^B5/B5^* testes sections (γH2AX: green and DAPI: gray). (B) Quantification of all sex body-positive cells per tubule (*Topbp1^+/+^* number of cells counted = 781, number of tubules counted = 28, number of mice = 3; *Topbp1^B5/B5^* number of cells counted = 967, number of tubules counted = 28, number of mice = 3). (C) Quantification of elongated-sex body cells per tubule (*Topbp1^+/+^* number of cells counted = 19 and 762 normal round/oval-sex bodies cells, number of tubules counted = 28, number of mice = 3; *Topbp1^B5/B5^* number of cells counted = 45 and 922 normal round/oval-sex bodies cells, number of tubules counted = 28, number of mice = 3).

(b) Regarding the 10X data, the finding that expression of some XY genes is elevated and others are not is also consistent with a "partial" phenotype (some cells have normal XY bodies and MSCI, others fail in both). In Fig 6E, X expression looks to be elevated in B5 vs wt at all stages...if this were a maintenance issue, shouldn't it be equal to that in wt and then elevate later?

We understand the point raised by the reviewer, however we do not favor the “partial” phenotype model because of the absence of any post-pachytene spermatocytes in the B5 mutant. If some cells had escaped the MSCI defect, we would expect to detect cells progressing further in meiosis. Because we cannot rule out completely the possibility of a subtle disruption in XY silencing initiation, we decided to better emphasize this point in the discussion (lines 391-394).

In Figure 6E, the X-linked genes were normalized against chromosome 9-linked genes. The normalization against pre-leptotene was done for the results displayed on Figure 7, in which we demonstrate the maintenance issue. Furthermore, for the 10X analysis, while the same number of cells were loaded for wild-type and mutant, the composition of cells varied between these two samples. Despite the fact that very few “spermatocyte 3” cells were detected in the mutant, those cells displayed much higher X-linked gene expression than the wild-type spermatocyte 3 cells.

1. How is the quantitation showing impaired localization of select markers (e.g. SETX) normalized? How do we know that the antibody staining simply didn't work as well on the mutant slides?

The quantification showing impaired localization of the selected markers such as SETX was done as described by Sims, et al. 2022 and Adams, et al. 2018. In brief, the green signal was measured along (XY cores) or across (XY DNA loops) the X and Y chromosomes and normalized against the analogous signal on the autosomal chromosomes. The possibility that the antibody simply did not work as well on the mutant is unlikely since multiple biological replicates were performed and we reproducibly followed standard practices in the field for meiotic spreads staining, imaging, and quantification. We also note that our findings published in Sims et al, 2022 show that ATR inhibition strongly impairs SETX localization to the sex body, further substantiating our claim that signaling via ATR-TOPBP1 controls SETX.

1. Is testis TOPBP1 protein expression reduced in the B5 mutant?

TOPBP1 protein abundance in the B5 mutant is reduced in lysates from whole testis, measured via western blot. We did not detect a significant reduction in TOPBP1 signal intensity measured by immunofluorescence in pachytene spreads of the B5 mutant.

1. 10X analysis: how were the genes on the y-axis in Figure 6-figure supplement 13 arranged? Is this by location on the X chromosome?

These genes were sorted by location across the chromosome X.

1. The final analyses in Fig 7: X-genes are subdivided based on their behavior (up, down, unchanged). What isn't clear to me is whether the authors have considered the fact that there are global changes in gene expression during meiosis (very low in lep , zyg and early pach, then ramps up hugely from mid pach). In other words, is this normalized to autosomal gene expression?

For the final analysis in Fig7, the normalization was done by their expression at the pre-leptotene stage. Moreover, the analysis was made comparing X-linked gene behavior in Wild-type vs B5 mutant.

1. Again regarding the 10X analysis, my prediction would be that not ALL X and Y gene would increase in pach if MSCI were ablated...we should remember that XY genes have been subject to MSCI for some 160 million years of evolution, and this will mean that many enhancers that originally drove their expression prior to the evolution of MSCI will now be lost. This has been our experience: many XY genes aren't elevated at pach even in mutants in which MSCI is totally defective. I'd urge the authors to consider this possibility when they use XY gene expression patterns to diagnose the severity or timing of the MSCI phenotype. This could be a discussion point.

We greatly appreciate the reviewer’s suggestion and have added discussion about this point to lines 392400.

**Reviewer #2 (Public Review):**
Summary:This paper described the role of BRCT repeat 5 in TOPBP1, a DNA damage response protein, in the maintenance of meiotic sex chromosome inactivation (MSCI). By analyzing a Topbp1 mutant mouse with amino acid substitutions in BRCT repeat 5, the authors found reduced phosphorylation of a DNA/RNA helicase, Sentaxin, and decreased localization of the protein to the X-Y sex body in pachynema. Moreover, the authors also found decreased repression of several genes on the sex chromosomes in the male mice.Strengths:The works including phospho-proteomics and single-cell RNA sequencing with lots of data have been done with great care and most of the results are convincing.Weaknesses:One concern is that, although the Topbp1 mutant spermatocytes show very severe defects after the stage of late pachynema, the defect in the gene silencing in the sex body is relatively weak. It is a bit difficult to explain how such a weak mis regulation of the gene silencing in mice causes the complete loss of cells in the late stage of spermatogenesis.

We appreciate the reviewer’s comment. We note that even subtle mis-regulation of XY gene silencing has been reported to lead to significant loss of cells in late stage of prophase I (Ichijima et al., 2011; Modzelewski et al., 2012). Moreover, it is possible that some cells with drastic changes in X-gene expression were excluded from the downstream analysis due to high levels of mitochondrial gene expression (cells that were likely dying due to apoptosis). The exclusion of cells with high levels of mitochondrial gene expression is a common practice in downstream analysis of sc-RNA sequencing data.

**Reviewer #3 (Public Review):**
The work presented by Ascencao and coworkers aims to deepen into the process of sex chromosome inactivation during meiosis (MSCI) as a critical factor in the regulation of meiosis progression in male mammals. For this purpose, they have generated a transgenic mouse model in which a specific domain of TOPBP1 protein has been mutated, hampering the binding of a number of protein partners and interfering with the regulatory cascade initiated by ATR. Through the use of immunolocalization of an impressive number of markers of MSCI, phosphoproteomics and single cell RNA sequencing (scRNAseq), the authors are able to show that despite a proper morphological formation of the sex body and the incorporation of most canonical MSCI makers, sex chromosome-liked genes are reactivated at some point during pachytene and this triggers meiosis progression breakdown, likely due to a defective phosphorylation of the helicase SETX.The manuscript presents a clear advance in the understanding of MSCI and meiosis progression with two main strengths. First, the generation of a mouse model with a very uncommon phenotype. Second, the use of a vast methodological approach. The results are well presented and illustrated. Nevertheless, the discussion could be still a bit tuned by the inclusion of some ideas, and perhaps speculations, that have not been considered.

We appreciate the reviewer’s comment and have improved the discussion section addressing the points raised in the “recommendation For the Authors”.

**Reviewer #1 (Recommendations For The Authors):**
I don't have any additional points here
**Reviewer #2 (Recommendations For The Authors):**
The paper by Ascencao et al. describes a separation-in-function allele of TOPBP1 critical for DNA damage response (DDR) that confers a specific defect in XY sex chromosome inactivation during male mouse meiosis. The authors constructed a Topbp1 separation-of-function mouse by introducing amino acid substitutions in BRCT repeat 5 and found the mice with normal DDR response in mitosis and meiosis show male infertility. Topbp1(B5/B5) mice do not contain spermatocytes after diplonema, as a result, little spermatids/sperms. In the mice, most of the meiotic events in prophase I including chromosome synapsis and meiotic recombination as well as the formation of the sex body are normal. The detailed proteomic analysis revealed the reduced ATR-dependent phosphorylation of a DNA/RNA helicase, Sentaxin. And also single-cell RNA sequencing found that the expression of some of genes from sex chromosomes are not silenced well compared to the control. The works with lots of data have been done with great care and most of the results are convincing. One clear concern is that, although the authors nicely showed a defect in gene silencing in sex chromosomes in the Topbp1(B5/B5) mice, how a small defect in the gene silencing leads to the complete loss of diplotene spermatocytes remains unaddressed.Major points:Although the authors showed a change in the transcriptome in spermatocytes of Topbp1(B5/B5) male mice, the authors cannot explain the complete lack of spermatids in this mouse. Even the transcriptome seems not to provide a clue.1. Given that the TOPBP1-B5 protein cannot bind to both 53BP1 and BLM, it is interesting to check the localization of both proteins on meiotic chromosome spreads (in the case of 53BP1, the localization in MEFs with DNA damage).

We appreciate the reviewer’s comment. We have tried to stain BLM in meiotic spreads using several different antibodies, however we were not successful getting specific signals for BLM. In the case of 53BP1, we monitored its localization, and it was not significantly different from *Topbp1^+/^*^+^ meiotic spreads, please refer to Figure 4-supplement figure 1. While we appreciate the reviewer’s suggestion of looking at the localization of 53BP1 in MEFs with DNA damage, we opted not to perform the experiment because we have shown that 53BP1 can still bind the BRCT 1 and 2 domains of TOPBP1 as previously described (Bigot et al., 2019; Cescutti et al., 2010; Liu et al., 2017). Additionally, both male and female 53BP1 KO mice are fertile (Ward et al., 2003), thus the partial disruption in binding to 53BP1 that we observed in TOPBP1 B5 mutant is likely not causing the infertility phenotype.

1. A recent preprint by Fujiwara et al. (doi: https://doi.org/10.1101/2023.04.12.536672) showed the accumulation of R-loops in spermatocyte spreads in Senataxin knockout mice. The authors may check the R-loop on the sex body in Topbp1-B5 mice.

We thank the reviewer for the suggestion. We have tried several protocols to stain R-loops (including the protocol used in the paper mentioned above) but were not successful.

1. The authors need to check the protein level (and band shift) of Senataxin in the testis by western blotting analysis.

We have tried several SETX antibodies, and none worked for western blot analysis.

1. If possible, the authors can see any protein interaction between TOPBP1 and Senataxin.

We appreciate the suggestion, and we will investigate this interaction in future work.

1. The authors need to check the statistics in the paper.

(1) It is better to show actual P-values in the case of "ns".

P-values were added to the respective figure legends.

(2) In focus counting such as Figures 3D, G, H, 4B, D, F, H, 5E, and F (and in Supplemental Figures), please indicate how many spreads were counted in each mouse. Moreover, the distribution of focus numbers and intensity of fluorescence are not parametric (not normal distribution). It is better to use a non-parametric method such as Mann-Whitney's U test.

We appreciate the reviewer's comment and upon consulting with a Statistician at Cornell Statistical Consulting Unit (CSCU), we were advised to use a linear mixed effect model to take into account the variability in cells within each mouse when comparing mice between groups (*Topbp1^+/+^* vs *Topbp1^B5/B5^*). We then reanalyzed all quantified meiotic spreads using this mixed effect model, and the p-value, number of mice, and number of cells counted for each group are displayed in the respective figure legends. Upon going through all the quantified meiotic spreads, we realized a minor error in one of the previous data points related to SETX staining in *Topbp1^+/+^* and have fixed it. Using the previous quantification data and the new stats analysis the p-value for cores was 0.5598 and *p-value for loops was 0.0273. Now using the correct values and the new stats analysis the p-value for cores is 0.5987 and *p-value for loops is 0.0452. The correction did not change the conclusion of this data and is now displayed in the new Figure 5. We also realized a mistake in the ATR quantification when the spreadsheet was moved from excel to Graphpad. Using the previous quantification and the new stats analysis the p-value for cores was 0.2451 and p-value for loops was 0.8933. Now using the correct values and the new stats analysis the p-value for cores is 0.4068 and p-value for loops is 0.9396. The correction did not change the conclusion of this data and is now displayed in the new Figure 4. Moreover, we realized that we used n = 8 (n = number of mice) for MDC1 quantification and n = 2 for pCHK1_S345, instead of n = 3 as shown in the preprint version of the manuscript. Corrected values were added to their respective figures and figure legends.

(3) From Figures 6E, 7B, and 7C, the authors conclude the difference in the expression profile between wild type and Topbp1(B5) spermatocytes. It is better to show P-values for the comparison. Particularly, in Figure 7C, Xiap expression kinetics look similar between wild type and the mutant.

We have added p-values to figures 6E and 7B and their respective figures or figure legends.

In figure 7C, we now recognize that the Δ could have been misleading as we meant to compare Wild-type SP2 to Wild-type SP3 and Mutant SP2 to SP3; and not comparing Wild-type SP3 to Mutant SP3.Therefore, the Δ was excluded from Figure 7C. For the comparisons between expression levels of SP2 and SP3, it is challenging to calculate p-values for a single gene since these cells have started X-gene silencing and expression values are very low. Meaningful p-values for the comparisons between Wildtype SP3 to Mutant SP3 can be visualized in Figure 7B, where the comparison is based on number of genes instead of expression levels of each gene.

Minor comments:1. Line 34: SPO11 is NOT a nuclease. Just delete it.

It has been deleted (see line 34).

1. Line 71, a protein: Is this protein ATR? Is so, please write it. If not, please give the name of the protein.

In line 71 (now lines 79-80), we refer to TOPBP1-interacting proteins in general since many of these interactions happen through a phosphorylation in the TOPBP1’s interactor. This is the case for BLM, 53BP1, FANCJ, and RAD9. ATR interacts with TOPBP1 through TOPBP1’s AAD domain and this is not a phospho-mediated interaction. We restructured the sentence for clarity.

1. In the Introduction, the authors often refer to a review by Cimprich and Cortez (2008) in various places. It is better to cite an original paper or the other an appropriate review.

We have accepted the reviewer’s suggestion and added original papers when appropriate.

1. Line 143-145: The authors generated eight charge reversal point mutations in the BRCT domain 5 of TOPBP1. If possible, it is helpful to mention the logic to generate these substitutions and also why BRCT domain 5, is not other domains.

We generated eight charge reversal point mutations to abrogate all possible phospho-dependent interactions and avoid potential residual interactions. We have mutated other BRCT domains as well, which will be published separately.

1. Line 174 (and Figure 2E): RPA should be either RPA2 or RPA32.

Corrected (it is RPA2).

1. Figure 5C-F: Please explain in more detail how the authors quantified the SETX signals. Why the two results are different?

The quantification was done as described by Sims, et al. 2022, yielding separate data for XY cores and DNA loops. In brief, the green signal was measured along (XY cores) or across (XY DNA loops) the X and Y chromosomes. Signals were normalized by the signal in the autosomal chromosomes.

**Reviewer #3 (Recommendations For The Authors):**
I have no major criticisms, but I include a list of comments and suggestions (some of them conceptual, and disputable) that could help the authors to improve some parts of the manuscript.1. Line 52: I realize that the term protein "sequestration" (used in many instances along the manuscript) has been widespread in the literature related to MSCI in the last years. While this might be a cool way to describe the dynamics of proteins accumulating in the sex body, this reviewer considers this term is totally inappropriate. It is confusing and introduces at least to mistakes to the fact of protein accumulation in the sex body. First, it seems to indicate that once trapped in the sex body, proteins are incapable of leaving it, which might be completely wrong (histone replacement refutes this idea). Second, it is suggested that DDR proteins are attracted by the sex body and cannot remain associated to autosomes even if DNA repair has not been completed. This has also been demonstrated to be incorrect (see for example PDMI 19714216). Moreover, DDR proteins can associate de novo to chromosomes if needed, for instance upon DNA damage caused by chemicals or irradiation. Thus, I suggest that the use of "sequestration" should be evaluated more critically, evaluating the misleading ideas that are subjacent to this term. The use of protein "accumulation" is much more objective and descriptive of the real facts.

We thank the reviewer’s suggestion and have addressed it in lines 52, 97 and 324.

1. Line 88: Just as a deference to the original ideas, it would be nice to acknowledge that the inactivation of sex chromosomes and the formation of a sex body in mouse meiosis was described more than 50 years ago (PDMI 5833946; 4854664). Likewise, the ideas about the sequential achievement and reinforcement of MSCI during pachytene have been developed during the last 20 years, far before the recent reports cited in the manuscript. Citations to these "old fashion" works would be great.

We appreciate the reviewer’s suggestion and have addressed it in line 86.

1. Line 90. Please, take into consideration that such a strong effect on meiosis progression occurs mainly in some knockout mice models and that in many other models (including hybrid mice models from natural populations) autosomal regions can remain unsynapsed and accumulate DDR proteins without impairing meiosis. In other mammalian species, meiosis is even more permissive to these MSUC phenomena.

We appreciate the reviewer’s suggestion and have addressed it at line 88.

1. Line 211: The differences in the abundance of MLH1 and MLH3 are remarkable. If these two proteins are supposed to form a heterodimer leading to crossover formation, then the increase of only MLH1 might be related to a different process, not leading to crossover (even not class II ones).

We agree with the reviewer’s comment and have included this point in the discussion (lines 491- 497).

1. Line 217: I have some doubts about the results presented in Figure 3-figure supplement 4. First, it is not clear to me how the represented cells counts were performed. Each spot is supposed to represent cell counts in a single individual, but how many cells were counted per individual? The proportion of cells could be a better indicator. Second, some B5/B5 individuals' counts were close to the ones displayed in the wild type. Did mutant animals show a high divergence compared to each other? It could be great to have each individual data displayed in a pie chart, and not only the aggregated data.

We have now addressed this in the Figure 3—figure supplement 4 legend. Each dot in the graph represents the sum of cells counted for each individual. We counted cells from 8 mice for each, *Topbp1^+/+^* and *Topbp1^B5/B5^*.

Here we summarize the total cells counted per individual:

**Author response table 1. sa4table1:** 

Mouse	Topbpl	Topbpl
1	26	43
2	46	72
3	43	84
4	79	64
5	37	29
6	48	52
7	74	67
8	52	8

1. Line 222: The data on 53BP1 deserve further attention. On the one side, from the analysis presented in Figure 4-figure supplement 1, it seems that 53BP1 tends to show a lower intensity in Topbp1B5/B5 mice. Since only 2 mice were analyzed, while for most of the other proteins 3-8 animals were studied, I suggest increasing the number of animals analyzed for 53BP1 localization, to test if this slight difference turns significant. This is relevant since: (1) the association of 53BP1 protein in somatic cells was clearly affected, and (2) 53BP1 is one of the last MSCI markers incorporated to the sex body at mid-late pachytene. These results should be moved to the main text and not appear as supplementary data. On the other hand, if no differences were to be found in meiosis, compared to somatic cells, how do authors explain these differences? Would 53BP1 have another partner at the sex body apart from TOPBP1? Could TOPBP1 have other BRCT domains (apart from domain 5) able to bind 53BP1?

We appreciate the reviewer’s suggestion; however, we had an issue with 53BP1 antibody. We analyzed 2 mice and needed to re-order the antibody. This antibody was backordered for almost one year, and when we finally received the order, the company had changed the clone for this antibody, and it no longer worked for meiotic spreads. In somatic cells, we see in HEK-293T a partial disruption in the binding to TOPBP1 B5 through IP-MS and IP-Western blot. The disruption is only partial due to the binding of 53BP1 to other domains in TOPBP1 such as BRCT 1 and 2 (Bigot et al., 2019; Cescutti et al., 2010; Liu et al., 2017). However, in assays in which we would expect a phenotypic response caused by impaired 53BP1, we did not see any effect, such as survival after IR (using the mice) and survival after phleomycin challenge (using Mefs). Moreover, 53BP1 KO mice, males and females, are fertile (Ward et al., 2003) so, the partial disruption in binding to 53BP1 that we observed in TOPBP1 B5 mutant is likely not causing the infertility phenotype.

1. Line 250: I do not understand what is represented in Figure 5A. Why did the author mix two different experiments (differences in phosphoprotein abundance in B5/B5 compared to wild type and the interference of ATR with AZ20)?

To account for the differences in cell population observed in the whole testis between *Topbp1^+/+^* and *Topbp1^B5/B5^*, and to know exactly which phosphorylation changes were due to disruption in the ATR signaling and not pleiotropic effects, we combined two different phosphoproteomes: One phosphoproteome from the comparison between *Topbp1^+/+^* and *Topbp1^B5/B5^* and another one from the comparison between Vehicle or ATR inhibitor-treated mice. By utilizing this approach, we only consider hits that were disrupted in both analyses. A similar method was used by Sims et.al, 2022 (Sims et al., 2022).

1. It is not clearly explained what is represented in Figure 6B. There is no explanation in the text or the figure legend. Do this represent the difference between scRNAseq in control and Topbp1B5/B5? If so, please, clarify.

We thank the reviewer’s comment and have addressed it in the legend of Figure 6B.

1. Line 342 and following. The authors describe a decrease of gene silencing. The use of two negative concepts is always confusing and results in the conversion to a positive one. I suggest considering the possibility of just talking about increase of gene expression, in order to make the message clearer.

We appreciate the reviewer’s point here, but it is important to note that the phenomenon disrupted in our mutants is MSCI, which is by definition a gene silencing mechanism. This phenotype is not as simple as “increased gene expression”, it is the removal of a mechanism that is a key feature of prophase I. Thus, because we are focusing on the mechanism of MSCI, it is crucial to maintain this (albeit unusual) terminology.

1. As for the classification of spermatocytes into 9 categories, I am curious about which spermatocytes are included in each of these categories. For instance, from cytology it seems that in Topbp1B5/B5 mice, spermatocytes are able to reach mid-late pachytene. However, in the spermatocyte categories established by scRNAseq they only reach class 3. Therefore, which are the populations included in the remaining 6 classes of spermatocytes? Do authors have any morphological correlation to these scRNAseq categories? Is it possible that in this mutant morphological advance of meiosis and gene expression profiles are uncoupled?

The clustering of cells to a specific group is based on RNA expression, which does not always match cytological features. Moreover, during the analysis, cells with high expression of mitochondrial genes are excluded (these are dying cells that do not pass the quality control). Thus, while *Topbp1^B5/B5^* reaches a mid-late-pachytene stage according to cytological analyses, in the single-cell RNA seq analysis we could only detect one pachytene stage. The other 6 remaining categories of spermatocytes can be classified according to their best-fit profile of gene expression. For that, we use the classification described by Chen et al., 2018 and Lau et al.,2020. Spermatocytes 3-5 = Pachytene, Spermatocytes 6-7 = Diplotene, Spermatocytes 8-9 = secondary spermatocytes (metaphase I/II). The gene markers used for this classification are displayed in Author response image 2.

**Author response image 2. sa4fig2:** Genes used as markers of spermatocytes captured in the scRNAseq analysis. Violin plots display the distribution of cells expressing Gm960 (Leptotene marker), Meiob(Leptotene/Zygotene marker), Psma8 (Pachytene marker), Pwill1 (Pachytene marker), Pou5f2 (Diplotene marker), and Ccna1 (Secondary Spermatocytes marker).

1. Figure 6E shows that overexpression of X-linked genes is not a feature of spermatocytes but it is initiated in spermatogonia. This fact has not been properly stated in the text and perhaps not sufficiently highlighted.

We noticed subtle changes during the spermatogonia stage and have addressed the reviewer’s comment in lines 317-322, however the downstream analyses related to a defect in X-gene silencing maintenance displayed in Figure 7 were done based on normalization of gene expression to its respective pre-leptotene stage.

1. Figure 6-figure supplement 13 shows that some X-linked genes are more expressed in Topbp1B5/B5 compared to control mice. In the figure it can be observed that many genes accumulate at the bottom of the graph. Does this have any correlation to the location of these genes along the X chromosome, for instance near or within the PAR? This could correlate with the defects in γH2AX accumulation at this region.

These are the locations along the chromosome. Only the bottom 5 rows are within the PAR region, so this accumulation is not within the PAR region specifically. The bottom tenth of the genes in the heatmap correspond to roughly a 17 Mb region.

1. The authors only analyzed the overexpression of genes located on the X chromosome. It would be interesting to show the behavior of Y-linked genes as well.

The coverage of Y-linked genes was not very high and that is why we have not shown the results in the paper. However, the results for Y-linked genes were similar to the X-linked genes and can be visualized in Author response image 3.

**Author response image 3. sa4fig3:** Single cell RNAseq reveals that *Topbp1^B5/B5^* spermatocytes initiate MSCI but fail to promote full silencing of Y chromosome-linked genes. Violin plot displaying the ratio of the average expression of Y chromosome genes by the average expression of chromosome 9 genes at different stages of spermatogenesis for *Topbp1^+/+^* and *Topbp1^B5/B5^* cells.

1. Line 425: Authors indicate that it is not known if association of TOPBP1 and BLM, 53BP1 or other proteins is disrupted in Topbp1B5/B5 spermatocytes. Could these experiments be performed in the testis, as they were in somatic cells?

The cellular composition in *Topbp1^+/+^* and *Topbp1^B5/B5^* testes is very different so it would not be a fair comparison. While we have tried to isolate pachytene cells to perform these experiments, we were successful only when using *Topbp1^+/+^* but not *Topbp1^B5/B5^*, likely due to the extremely small size of the mutant testis.

1. Line 455 and following. I find that the discussion about the role of SETX is not completely clear. It seems that a failure of SETX function could result in defective or no transcription, as a consequence of the impossibility to resolve RNA-DNA hybrid molecules. Therefore, should impairment of SETX lead to reduced or enhanced transcription? Please clarify. On the other hand, this defect in SETX function should affect the whole genome, and not only sex chromosomes. Do authors have any clues about this broad effect?

We thank the reviewer’s comment and have expanded on discussion in lines 470-474. While we agree with the reviewer’s point that an impairment on SETX should affect the whole genome, however, during pachytene stage, SETX is mostly localized to the sex body. The *Topbp1^B5/B5^* shows a specific defect in X and Y silencing maintenance during pachytene stage, thus we hypothesized that an impairment in SETX localization during pachytene should especially impair the X and Y chromosomes.

1. As a general comment to the discussion section, I think authors could extend into some specific ideas or speculations. It is shocking that sex chromosome-linked genes are able to escape silencing without dismantling the complex (almost complete) MSCI response in the Topbp1 mutant (although perhaps this is not so surprising considering the high number of escapees reported in the inactivated X chromosome in female somatic cells).How to explain this paradox? One possibility (which would make a real breakthrough) is that the expression of sex chromosome-linked genes represents a regulated response to meiotic defects, and not just an unfortunate consequence of a defective MSCI. Thus, MSCI might be somehow irrelevant to prevent the execution of this sex chromosome-based program to stop meiosis progression when needed. The fact that this regulated activation was never proposed is perhaps due to the fact that most of the meiosis mutants characterized so far are unable to reach the stage at which MSCI is properly established, which is the most remarkable difference with the Topbp1 mutant studied here.Although naïve, the critical point for the activation of this sex chromosome-based program seems to depend simply on the transcription of Zfy1 and Zfy2 (encoding for transcription factors). The signaling cascades up and downstream these genes are the real mystery, awaiting further studies.

We thank the very interesting point raised by the reviewer. Our interpretation of the data is that X and Y silencing being a dynamic process requires an initiation step and a maintenance step driven/controlled by the DDR machinery, and that *Topbp1^B5/B5^* shows a grossly normal initiation of X and Y silencing but fails on maintain MSCI. Moreover, the expression of *Zfy1* and *Zfy2* have been previously demonstrated as enough to trigger cell death (Royo et al., 2010; Vernet et al., 2016), and *Topbp1^B5/B5^* cells show increased expression of these genes. However, we do not exclude the very interesting possibility, raised by the reviewer, that the expression of XY-linked genes represents a regulated response to meiotic defects to stop meiosis progression, leading to the cell death observed in *Topbp1^B5/B5^*, which makes the *Topbp1^B5/B5^* an unique model for these studies as most of the previous meiosis mutants are unable to reach the stage at which MSCI is properly established. We add discussion about this exciting point in lines 513-522.

1. Scale bars are impossible to read in Figures 1I and J, and are missing in all the other image figures. Please, correct.

We have addressed this in the new Figure 1. For figures displaying meiotic spreads, adding a scale bar is not a common practice in the field as these cells are swollen while being prepared.

1. Line 828. Since Paula Cohen is an author of the manuscript, it seems weird to acknowledge herself in this section.

Corrected.

References

Adams SR, Maezawa S, Alavattam KG, Abe H, Sakashita A, Shroder M, Broering TJ, Sroga Rios J, Thomas MA, Lin X, Price CM, Barski A, Andreassen PR, Namekawa SH. 2018. RNF8 and SCML2 cooperate to regulate ubiquitination and H3K27 acetylation for escape gene activation on the sex chromosomes. PLoS Genet 14. doi:10.1371/journal.pgen.1007233

Bigot N, Day M, Baldock RA, Watts FZ, Oliver AW, Pearl LH. 2019. Phosphorylation-mediated interactions with topbp1 couple 53bp1 and 9-1-1 to control the g1 DNA damage checkpoint. Elife 8:1–28.

Cescutti R, Negrini S, Kohzaki M, Halazonetis TD. 2010. TopBP1 functions with 53BP1 in the G1 DNA damage checkpoint. EMBO J 29:3723–3732.

Chen Y, Zheng Y, Gao Y, Lin Z, Yang S, Wang T, Wang Q, Xie N, Hua R, Liu M, Sha J, Griswold MD, Li J, Tang F, Tong M-H. 2018. Single-cell RNA-seq uncovers dynamic processes and critical regulators in mouse spermatogenesis. Cell Res 28:879–896.

Hirota T, Blakeley P, Sangrithi MN, Mahadevaiah SK, Encheva V, Snijders AP, ElInati E, Ojarikre OA, de Rooij DG, Niakan KK, Turner JMA. 2018. SETDB1 Links the Meiotic DNA Damage Response to Sex Chromosome Silencing in Mice. Dev Cell 47:645-659.e6.

Ichijima Y, Ichijima M, Lou Z, Nussenzweig A, Daniel Camerini-Otero R, Chen J, Andreassen PR, Namekawa SH. 2011. MDC1 directs chromosome-wide silencing of the sex chromosomes in male germ cells. Genes and Development 25:959–971.

Lau X, Munusamy P, Ng MJ, Sangrithi M. 2020. Single-Cell RNA Sequencing of the Cynomolgus Macaque Testis Reveals Conserved Transcriptional Profiles during Mammalian Spermatogenesis. Dev Cell 54:548-566.e7.

Liu Y, Cussiol JR, Dibitetto D, Sims JR, Twayana S, Weiss RS, Freire R, Marini F, Pellicioli A, Smolka MB. 2017. TOPBP1Dpb11 plays a conserved role in homologous recombination DNA repair through the coordinated recruitment of 53BP1Rad9. J Cell Biol 216:623–639.

Modzelewski AJ, Holmes RJ, Hilz S, Grimson A, Cohen PE. 2012. AGO4 regulates entry into meiosis and influences silencing of sex chromosomes in the male mouse germline. Dev Cell 23:251–264. Royo H, Polikiewicz G, Mahadevaiah SK, Prosser H, Mitchell M, Bradley A, De Rooij DG, Burgoyne PS, Turner JMA. 2010. Evidence that meiotic sex chromosome inactivation is essential for male fertility. Curr Biol 20:2117–2123.

Sims JR, Faça VM, Pereira C, Ascenção C, Comstock W, Badar J, Arroyo-Martinez GA, Freire R, Cohen PE, Weiss RS, Smolka MB. 2022. Phosphoproteomics of ATR signaling in mouse testes. Elife

1. doi:10.7554/eLife.68648

Vernet N, Mahadevaiah SK, de Rooij DG, Burgoyne PS, Ellis PJI. 2016. Zfy genes are required for efficient meiotic sex chromosome inactivation (MSCI) in spermatocytes. Hum Mol Genet 25:5300–5310.

Ward IM, Minn K, van Deursen J, Chen J. 2003. p53 Binding protein 53BP1 is required for DNA damage responses and tumor suppression in mice. Mol Cell Biol 23:2556–2563.

Yeo AJ, Becherel OJ, Luff JE, Graham ME, Richard D, Lavin MF. 2015. Senataxin controls meiotic silencing through ATR activation and chromatin remodeling. Cell Discovery 1. doi:10.1038/celldisc.2015.25